# Partial connectomes of labeled dopaminergic circuits reveal non-synaptic communication and axonal remodeling after exposure to cocaine

Gregg Wildenberg[1,2]*, Anastasia Sorokina[1,2], Jessica Koranda[1], Alexis Monical[3], Chad Heer[1], Mark Sheffield[1], Xiaoxi Zhuang[1], Daniel McGehee[3], Bobby Kasthuri[1,2]*

[1]Department of Neurobiology, University of Chicago, Chicago, United States; [2]Argonne National Laboratory, Lemont, United States; [3]Department of Anesthesia & Critical Care, University of Chicago, Chicago, United States

**Abstract** Dopaminergic (DA) neurons exert profound influences on behavior including addiction. However, how DA axons communicate with target neurons and how those communications change with drug exposure remains poorly understood. We leverage cell type-specific labeling with large volume serial electron microscopy to detail DA connections in the nucleus accumbens (NAc) of the mouse (*Mus musculus*) before and after exposure to cocaine. We find that individual DA axons contain different varicosity types based on their vesicle contents. Spatially ordering along individual axons further suggests that varicosity types are non-randomly organized. DA axon varicosities rarely make specific synapses (<2%, 6/410), but instead are more likely to form spinule-like structures (15%, 61/410) with neighboring neurons. Days after a brief exposure to cocaine, DA axons were extensively branched relative to controls, formed blind-ended 'bulbs' filled with mitochondria, and were surrounded by elaborated glia. Finally, mitochondrial lengths increased by ~2.2 times relative to control only in DA axons and NAc spiny dendrites after cocaine exposure. We conclude that DA axonal transmission is unlikely to be mediated via classical synapses in the NAc and that the major locus of anatomical plasticity of DA circuits after exposure to cocaine are large-scale axonal re-arrangements with correlated changes in mitochondria.

*For correspondence:
gwildenberg@uchicago.edu (GW);
bobby.kasthuri@gmail.com (BK)

**Competing interest:** The authors declare that no competing interests exist.

## Editor's evaluation

How dopaminergic neurons communicate with other neurons, whether via point-to-point contact involving classical synapses or through volume transmission, has remained controversial. By performing large-scale serial electron microscopy combined with genetic labelling methods, this study reveals that dopaminergic axonal varicosities lack features of classical synapses and that following exposure to cocaine dopaminergic axons undergo extensive remodeling. These findings provide major insights into the biology of dopaminergic axons and are of fundamental interest, and they also form a basis for understanding dopaminergic circuit changes associated with drugs of abuse.

## Introduction

The dopaminergic (DA) system, like many neuromodulatory systems of the brain, profoundly influences behaviors including learning risks and rewards, social cooperation, goal-directed behavior, and

decision making (*Alcaro et al., 2007*; *Eskenazi et al., 2021*; *Liu et al., 2021*). Moreover, alterations in DA circuitry in animal models and humans are a hallmark of addictive behaviors and are likely the result of changes in how DA neurons communicate with targets (*Berke and Hyman, 2000*). However, while much is known about the molecular composition and functional roles of DA neurons in normal behavior and addiction (*Alcaro et al., 2007*; *Beeler et al., 2009*; *Berke and Hyman, 2000*; *Liu and Kaeser, 2019*; *Liu et al., 2018*), much less is known about how DA neurons physically communicate with targets and how that communication changes with exposure to drugs of abuse.

There were several problems preventing progress. One is that, until recently, collecting and analyzing large volumes of brain tissue with electron microscopy (EM), the gold standard for delineating neuronal connections, had been difficult and laborious. Recent advances in automation of the data collection (*Hayworth et al., 2020*; *Kasthuri et al., 2015*; *Xu et al., 2017*; *Yin et al., 2020*) and algorithms to analyze the resulting terabytes of data have made such approaches more accessible (*Funke et al., 2019*; *Januszewski et al., 2018*; *Saalfeld et al., 2012*; *Turaga et al., 2010*). A second problem is that, unlike ionotropic excitatory or inhibitory connections, there is much less known about the ultrastructural characteristics of potential DA synapses, where they occur on neurons, or how frequently (*Liu et al., 2018*; *Rice and Cragg, 2008*). Thus, analyzing DA circuits has required specific labeling, either with immunohistochemistry (*Bérubé-Carrière et al., 2012*; *Moss and Bolam, 2008*; *Omelchenko and Sesack, 2009*; *Liu et al., 2018*) or with genetic targeting (*Dos Santos et al., 2018*; *Melchior et al., 2021*; *Mingote et al., 2019*; *Nasirova et al., 2021*; *Poulin et al., 2018*). However, fundamental questions remain unanswered: (1) how often do DA axons make synapses that appear like classic chemical synapses (i.e., with synaptic vesicles and post-synaptic densities [PSDs]), (2) where do DA synapses occur on target neurons, (3) what do DA axonal varicosities contain, and (4) are there signs of other physical interactions between DA axons and their targets?

Here, we combine recent advances in serial EM ('connectomics') (*Bae et al., 2018*; *Bates et al., 2020*; *Briggman et al., 2011*; *Helmstaedter et al., 2013*; *Karimi et al., 2020*; *Kasthuri et al., 2015*; *Morgan and Lichtman, 2020*) and genetic labeling approaches for EM (*Joesch et al., 2016*; *Martell et al., 2017*; *Sampathkumar et al., 2021b*; *Tsang et al., 2018*) to characterize the 3D architecture of dopamine axon wiring in the mouse (*Mus musculus*). Moreover, since DA pathways appear highly plastic (*Calabresi et al., 2007*; *Jay, 2003*; *Pignatelli and Bonci, 2015*; *Pignatelli et al., 2017*), especially with exposure or addiction to drugs of abuse (*Berke and Hyman, 2000*; *Lüscher, 2016*; *Nestler, 2001*; *Saal et al., 2003*; *Ungless et al., 2001*), we wondered whether connectomics could reveal potential structural changes of that plasticity. We focused on potential early changes after brief exposure to cocaine, testing the sensitivity of large volume EM for revealing acute changes. We labeled ventral tegmental area (VTA) dopamine neurons in two cocaine sensitized and three control mice using the genetically encoded pea peroxidase gene, Apex2, and created serial EM datasets of dopamine neurons in the VTA and their axons in the nucleus accumbens (NAc) (~0.5mm × 0.5 mm × 0.03 mm volumes for each brain region) using the ATUM approach (*Kasthuri et al., 2015*).

We found several novel aspects of DA axonal biology including a lack of ultrastructural evidence of synapses, axonal varicosities of different morphological types, and evidence of 'spinules', where the membranes of DA axonal varicosities interdigitated with the membranes of nearby excitatory and inhibitory axons as well as dendrites of resident neurons, reminiscent of spinules in other parts of the brain (*Petralia et al., 2015*; *Spacek and Harris, 2004*; *Tao-Cheng et al., 2009*; *Tarrant and Routtenberg, 1977*; *Westrum and Blackstad, 1962*). Brief cocaine exposure shows evidence of large-scale axonal re-arrangements: increases in axon branching and ~50% of DA axons contained branches ending in blind 'bulbs', reminiscent of retraction bulbs in the developing and injured brain (*Balice-Gordon et al., 1993*; *Bishop et al., 2004*; *Bixby, 1981*; *Canty et al., 2020*; *Ertürk et al., 2007*; *Johnson et al., 2013*; *Korneliussen and Jansen, 1976*; *Riley, 1981*), often surrounded by elaborated processes of nearby glia. Also, post-exposure, mitochondria in DA axons and putative targets were 2.2-fold longer but mitochondria were not changed in NAc glutamatergic axons or in VTA DA soma or DA dendrites. Overall, we conclude that connectomic tools can give new insights into DA axon biology and how brief exposure to cocaine impacts mesoaccumbens dopamine circuitry.

## Results

To selectively label mesoaccumbens dopamine neurons, we bilaterally injected adeno-associated virus (AAV) encoding either a Cre-dependent cytosolic or mitochondrial targeted pea peroxidase,

Apex2 (*Martell et al., 2017*; *Sampathkumar et al., 2021a*), into the VTA of five mice expressing Cre recombinase from the promoter of the dopamine transporter gene (DAT-CRE) (*Zhuang et al., 2005*; *Figure 1A*). The mouse line used targets Cre recombinase to the *Slc6a3* locus that encodes DAT to create the DAT-Cre knock-in strain. As a result, it disrupts one copy of DAT. Losing one copy of DAT is known to slightly reduce DA reuptake but heterozygote mice do not display any other phenotypes (*Giros et al., 1996*). In our cocaine experiments, the cocaine and saline groups have the same genotype to ensure comparisons are not confounded by strain differences.

Four weeks following AAV expression, mice were perfused, and brain slices were treated with 3,3′-diaminobenzidine (DAB) to convert Apex2 into a visible precipitate with binding affinity to osmium tetroxide (*Figure 1B*; *Joesch et al., 2016*; *Martell et al., 2017*). Brain sections with DAB precipitate localized to the appropriate brain regions were cut into smaller pieces surrounding the VTA and the medial shell of the NAc (*Figure 1B*, green boxes) and further processed for serial EM (*Hua et al., 2015*) (see Materials and methods). We focused our analysis on DA axons projecting to the medial shell of the NAc (referred to as simply, NAc) because: (1) It receives the majority of VTA projecting DA axons (*Beckstead et al., 1979*). (2) It is reported to undergo functional and structural plasticity in response to cocaine (*MacAskill et al., 2014*). (3) Areal landmarks allow for unambiguous identification of NAc across replicates (*Figure 1B*, example landmarks depicted with red ovals highlighting negative staining of the anterior commissure and nuclei along midline of the brain, and see Materials and methods). We first collected single 2D EM sections to confirm that both cytosolic- and mitochondrial-localized Apex2 unambiguously labeled DA neurons at their soma (*Figure 1C–D*, left panels, red arrow), dendrites (*Figure 1C–D*, middle panels, cyan arrow), and small axonal processes several millimeters away from the injection site (*Figure 1C–D*, right panel, yellow arrow). We found no evidence of labeled soma in any other brain region, including the NAc, suggesting that the viral strategy worked, targeting only genetically specified DA axons in an anterograde manner. After confirming appropriate Apex2 expression for all mice used in this study, ~2000, 40-nm-thick sections were collected from both the VTA and NAc, and volumes of ~0.5 mm × 0.5 mm × 0.03 mm were imaged by serial EM.

We first described ultrastructural features of putative DA contacts with target neurons (i.e., what, if any, types of synapses do DA axons make?). We characterized DA axonal varicosities (see Materials and methods) as the most likely location along axons for evidence of synaptic contacts in DA axons in 'control' mice not subject to any of the behavioral experiments used in the cocaine sensitization experiments described below. We used mito-Apex2 to clearly visualize ultrastructural features (e.g., vesicle clouds, PSD), which might be obscured by cytoplasmic Apex2 expression. We reconstructed 410 varicosities on 75 individual DA axonal fragments in one mouse (age = p105, male). We defined varicosities as regions where the DA axon diameter increased ~3-fold relative to the thinnest portion of the axon. Broadly, varicosities were of four types based on their contents: (1) empty (38%: 156/410), (2) several small vesicles (25%: 101/410; 48 ± 1.7 nm in diameter), (3) few, large vesicles (19%: 78/410; 133 ± 5.5 nm in diameter), or (4) mixture of large and small vesicles (18%: 75/410) (*Figure 2A*). *Figure 2B* shows representative examples of each varicosity type on a single axon colored to correspond to the pie chart of all varicosity distributions in *Figure 2A* (red arrows or arrowheads point to large and small vesicles, respectively, white arrow pointing to an empty varicosity, asterisks marks Apex2+ mitochondria). As an additional metric to distinguish between these varicosity types, we calculated the distribution of number of vesicles for each type (vesicle number mean ± SEM: small = 30.1 ± 3.2, large = 9.2 ± 0.8, large/small total = 35.1 ± 5.5, large/small: large only = 6.9 ± 0.9, large/small: small only = 28.2 ± 4.8). Although individual axons often contained more than one type of varicosity (*Figure 2C*), we found across many axons, long stretches of a particular varicosity type (*Figure 3A*), suggesting a preference of individual axons for a specific type of varicosity. Thus, we asked whether such distributions along individual axons were consistent with random sampling of varicosity type, weighted by their population frequencies (i.e., *Figure 2A*). We performed a Monte Carlo simulation (*Rubinstein and Kroese, 2008*) such that, for every DA axon with three or more varicosities reconstructed from the real dataset, a DA axon was simulated with the exact number of varicosities/axon so that all axons in the real dataset had an exact replicate in the simulation. For every trial of the simulation (100,000 trials), we randomly assigned a varicosity type, based on their population frequency, to each varicosity of a simulated DA axon. Finally, we asked how often in the 100,000 trials do simulated axons have long stretches of a particular varicosity (i.e., how many simulated axons have more than one, more than two, or more than three, etc. of Type I, II, III, or IV varicosity). We

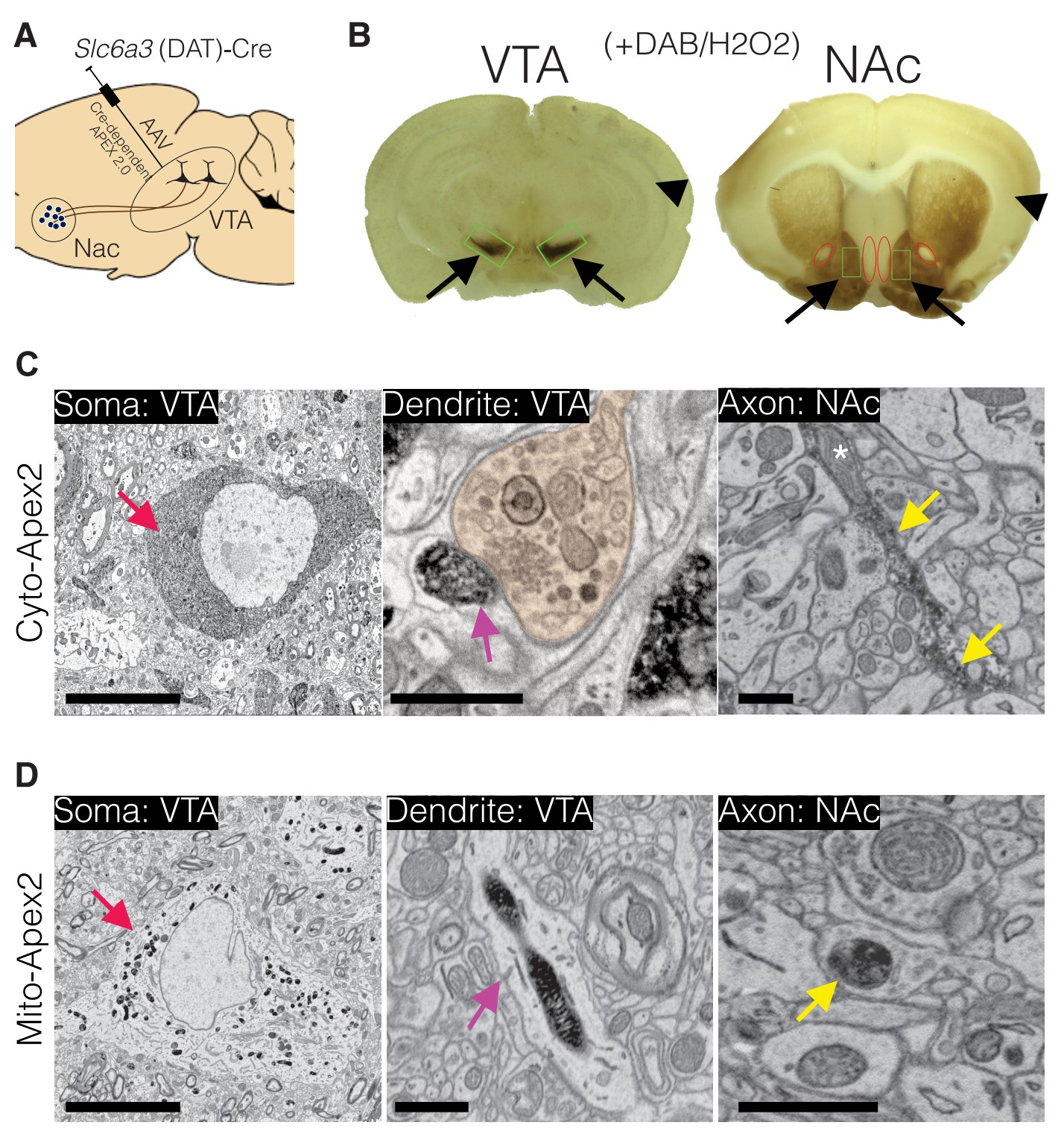

**Figure 1.** Experimental design for dopamine connectomics. (**A**) Adeno-associated viruses (AAVs) expressing Cre-dependent Apex2 were bilaterally injected into the ventral tegmental area (VTA) of transgenic mice expressing Cre in dopamine transporter positive neurons (*Slc6a3* (DAT)-CRE). (**B**) Approximately 4 weeks after AAV injections, vibratome sections (~300 μm thick) show strong Apex2 labeling in VTA and nucleus accumbens (NAc) after staining with 3'3'-diaminobenzidene (DAB) and hydrogen peroxide ($H_2O_2$) before electron microscopy (EM) processing (see Materials and methods). Black arrows point to an Apex2-positive region and black arrowhead points to an Apex2-negative region. Green rectangles highlight the VTA and medial shell of the NAc region dissected out and processed for EM. Red ovals highlight areal landmarks to ensure the same region was dissected across

*Figure 1 continued on next page*

*Figure 1 continued*

all animals. (**C–D**) Representative EM images of cytoplasmic (C, top row) and mitochondrial (D, bottom row) Apex2+ DA neurons. Left panel: Apex2 soma in the VTA (red arrows). Middle panel: Apex2 top panel shows a DA dendritic spine forming a synapse (purple arrow) with presynaptic bouton (orange) in the VTA, and the bottom panel shows a narrow DA dendrite expressing mitochondrial Apex. Right panel: Apex2 axon in the NAc with narrow (yellow arrow) and thick varicosities (yellow arrowhead). Cytosolic Apex2 does not obscure mitochondria (top panel, asterisk), and mitochondrial Apex2 (bottom) only fills up mitochondria. Scale bar = C,D, soma = 10 µm, dendrite and axon = 1 µm.

found that actual DA axons were far more likely to contain long stretches of all varicosity types, relative to the simulated population (*Figure 3B*, and see *Source code 1* ). For example, Type II, III, and IV varicosities appeared multiple times along single axons (e.g., >3 instances), which rarely occurred in over 100,000 trials of the simulation (see *Figure 3—source data 1* for all p-values). We next asked whether the inter-varicosity distance was correlated with different varicosity types as a potential clue for varicosity types being organized along an axon depending on their vesicle contents. We found the distances between pairs of varicosities (n = 170 varicosities across 29 DA axons) fit a log normal distribution (*Figure 3—figure supplement 1A*) without any obvious signs of extremely long or short inter-varicosity distances clustering to a particular varicosity type. Finally, we scored for the presence or absence of mitochondria in each varicosity type to further ascertain whether mitochondria were uniquely associated with certain varicosity types. However, we found that for each varicosity type, there was an ~50% chance that a mitochondrion was also present, eliminating any obvious association between mitochondrial location and varicosity type (*Figure 3—figure supplement 1B*).

We conclude that individual DA axons have multiple instances of the same varicosity type which are unlikely to have occurred by chance and suggest that DA axons can be classified by the types of varicosities present along an axon.

In rare instances (6/410), we found clear ultrastructural evidence (i.e., clusters of vesicles with clear PSDs in the target neuron, parallel apposition of presynaptic axonal, and target neuron membranes) of DA axons making synapses on the soma and dendritic shaft of resident NAc neurons. *Figure 4* shows three examples of mito-Apex2+ DA axons forming synapses on soma and the shaft of resident NAc neurons. In each case, the DA axons varicosity had a prominent vesicle cloud localized around a PSD (red arrows) (see *Figure 4—figure supplements 1 and 2* for montage of DA synapses for more details). Finally, we examined whether unlabeled boutons, presumably from glutamatergic excitatory axons, also showed clear ultrastructural evidence of synapses. We returned to the same mito-Apex2 dataset and identified 100 axonal boutons containing a large vesicle cloud (i.e., 50 or more small vesicles) and asked whether we could identify an obvious PSD in membrane-to-membrane apposition to the bouton. We found that 100/100 presynaptic boutons were in close apposition to a post-synaptic dendritic spine or shaft with a stereotypically dark PSD staining, thus mitigating the possibility that the lack of observed DA synapses is due to our sample preparation or imaging approach. *Figure 4— figure supplement 3* depicts three representative examples of 'chemical' synapses used for this analysis. These results demonstrate both that our approach can detect rare DA synapses and also that the majority of DA varicosities show no evidence of widespread structural connections with other neurons.

Lastly, we asked whether DA axons physically interact in any other way with other neurons in the volume. We found in mito-Apex2 expressing DA axons that 15% (61/410) of varicosities contained membrane invaginations with either unlabeled axons or dendrites. Representative examples are shown as image montages and 3D reconstructions from both mito- and cyto-Apex2 datasets (*Figure 5* and *Figure 5—figure supplement 1*). Furthermore, these invaginations or 'contact points' were made primarily between DA axons and unlabeled axons, most anatomically similar to chemical synapse (CS) forming axons (e.g., glutamatergic or GABAergic axons) (83%, 49/59) and a smaller proportion with dendrites (17%, 10/59). The unlabeled CS axons that made invaginations into DA axons could be further classified into a cohort (~43%) making additional synapses on dendritic spines, suggesting they were excitatory and the rest, 57%, made synapses on dendritic shafts with little sign of a PSD, suggesting they were inhibitory. All dendrites with contact points with DA axons were 100% spiny. These invaginations closely resemble previously characterized structures called spinules which are believed to be post-synaptic projections into presynaptic terminals (*Petralia et al., 2015*; *Spacek and Harris, 2004*; *Tao-Cheng et al., 2009*; *Tarrant and Routtenberg, 1977*; *Westrum and Blackstad, 1962*) and have recently been shown to be induced by neuronal activity (*Tao-Cheng et al., 2009*; *Zaccard et al., 2020*) (see Discussion).

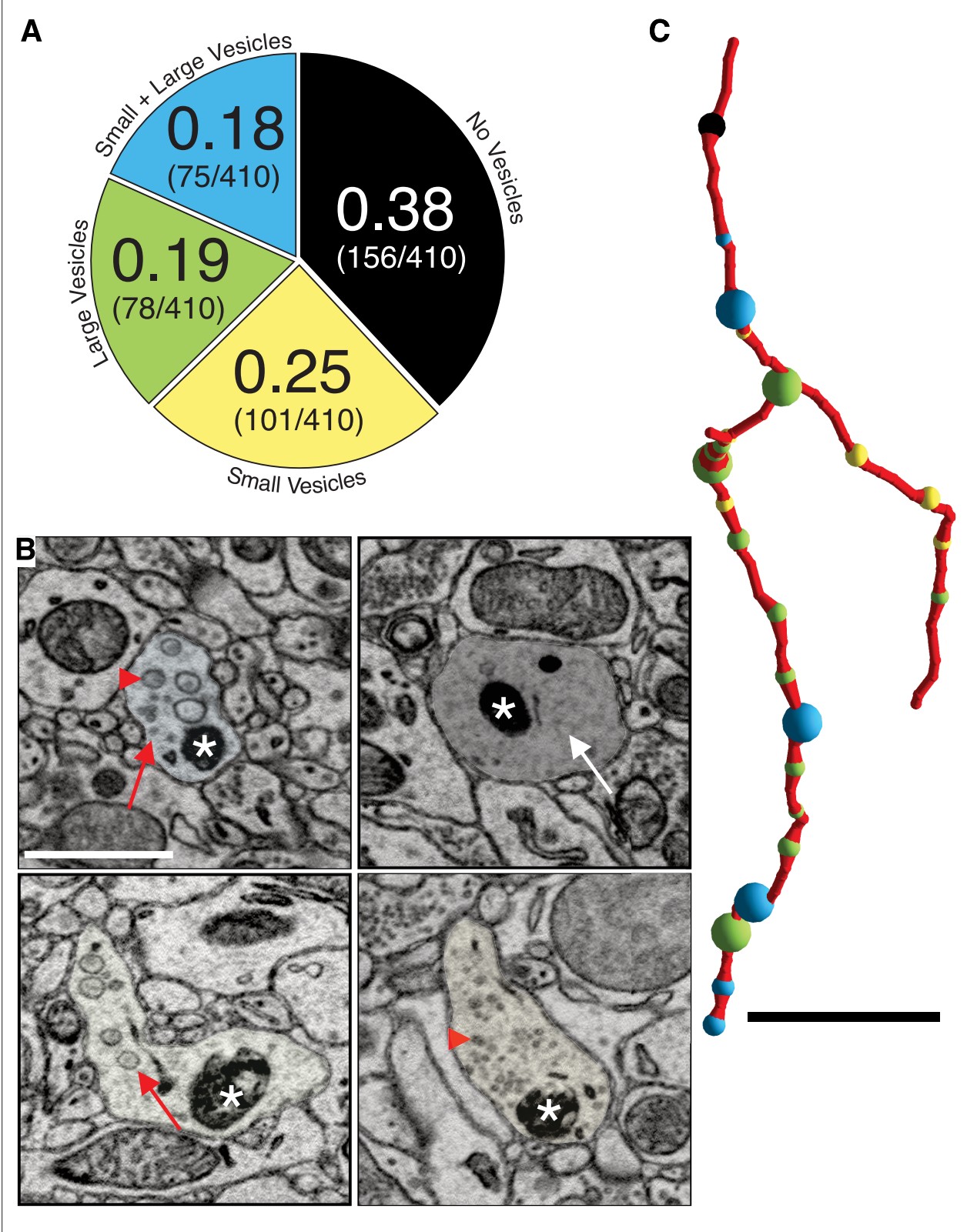

**Figure 2.** Dopaminergic (DA) axon varicosities are either empty or filled with different sized vesicles. (**A**) Pie chart showing ratio of each kind of DA varicosity across a population of DA axons (n = 409 varicosities scored across 75 axons, 1 mouse). (**B**) Single 2D images showing representative examples of each kind of DA axon varicosity with color overlay corresponding to each type (refer to pie chart in A). Red arrowheads point to small vesicles, red arrows point to large vesicles, and the white arrow points to an empty varicosity. Asterisks marks Apex2+ mitochondria. (**C**) Reconstruction of a single

*Figure 2 continued on next page*

*Figure 2 continued*

Mito-Apex2 DA axon with each class of varicosity marked as a differently colored spheres: black = no vesicles, yellow = small vesicles, green = large vesicles, and blue = small and large vesicle-filled varicosity. Spheres are scaled to the size of the bouton. Scale bar (**B**) = 1 µm, (**C**) = 10 µm.

Given this basic appreciation of DA circuits, we next asked whether connectomic datasets could reveal changes in the physical circuitry of DA axons after exposure to cocaine. Two DAT-CRE mice expressing cytosolic Apex2 after AAV injections were subjected to a cocaine sensitization and associative conditioning protocol previously shown to induce morphological changes in the mesoaccumbens dopamine pathway (*Beeler et al., 2009*; *Li et al., 2004*; *Singer et al., 2009*) and compared to two controls injected with equivalent volumes of saline and exposed to the same associative conditioning protocol. Briefly, mice were given one daily intraperitoneal (IP) injection of either cocaine (10 mg/kg, n = 2) or an equivalent volume of saline (n = 2) every other day for a total of four injections (*Figure 6— figure supplement 1A*) and assessed for locomotor activity in a novel environment. Consistent with prior reports, cocaine-treated mice showed an increase in daily activity relative to the control with each successive treatment (*Figure 6—figure supplement 1B*, C). After the final injection, mice went through a standard 4-day abstinence period, before being processed for Apex2 staining and large volume serial EM. A 4-day abstinence period was chosen to minimize potential acute effects of cocaine on dopamine circuitry, focusing this study on potentially more long-term changes caused by cocaine sensitization (*Benuck et al., 1987*; *Eipper-Mains et al., 2013*; *Grimm et al., 2001*; *Hollander and Carelli, 2007*; *Nestler, 2001*; *Parsons et al., 1991*).

We first analyzed the trajectories of axons in 20 nm (x, y) resolution EM volumes from NAc datasets of two control (+saline) and two cocaine-treated (+cocaine) mice expressing cyto-Apex2. Cyto-Apex2 was used for this analysis because it allowed us to follow thin axons at lower resolutions. We traced 85 cyto-Apex2+ DA axons (44 from controls, 41 from cocaine treated, total length of 5192.8 µm) and immediately found that DA axons were highly branched in cocaine-treated mice relative to controls (*Figure 6A*; +saline = blue axons, +cocaine = red axons). Across labeled DA axons, there was an average 4-fold increase in branch number for similar axon lengths with exposure to cocaine (*Figure 6B*; mean ± SEM branch number/µm length of axon: +saline, 0.01 ± 0.002, n = 44 axons, two mice; +cocaine, 0.04 ± 0.005, n = 41 axons, two mice. p = 3.96e-8). This increased branching was accompanied by an increased number of invaginating contact points (i.e., 'spinules') (see *Figure 6A*, image insets and *Figure 6—figure supplement 2* for examples of contact points identified in 20 nm resolution datasets) in cocaine exposed animals as described above in *Figures 2 and 5*, but the number of contacts per length was similar. (*Figure 6C*; mean ± SEM number of contacts points (c.p.)/ µm length of axon: +saline, 0.2 ± 0.07, n = 240 contact points across 14 axons, two mice; +cocaine, 0.2 ± 0.03, n = 142 contact points across nine axons, two mice. p = 0.90.) Finally, we did not see an obvious difference in which kind of neuronal process formed invaginating contact points (e.g., axo-axonic versus axo-dendritic) (*Figure 6—figure supplement 3*: control: 83% (49/59) axo-axonic, 17% (10/59) axo-dendritic. n = 59 contact points across 19 DA axons; cocaine: 77% (56/73) axo-axonic, 23% (17/73) axo-dendritic, n = 73 contact points across 20 DA axons). These results suggest that while as axons make new branches, they also form new contact points with surrounding neurons.

The second obvious feature of DA axons exposed to cocaine was the occurrence of large axonal swellings or bulbs (*Figure 7*). The swellings were large (mean ± SEM diameter: 2.2 ± 0.3 µm, n = 23), significantly larger than varicosities in control animals (mean ± SEM diameter: 0.4 ± 0.02 µm, n = 118 varicosities) and at times reaching the size of neuronal soma (*Figure 7A*). These 'bulbs' were common in axons (~56%, 17/30 axons) in two cocaine exposed animals and we did not see a single example in DA axons from two control animals (0/29 axons), suggesting that Apex2 expression alone does not cause these swellings (*Figure 7B*; mean ± SEM swellings/µm length of axon: +saline, 0.00 ± 0.0, n = 29 axons, two mice; +cocaine, 0.04 ± 0.02, n = 30 axons, two mice. p = 1.7e-5). *Figure 7C* shows reconstructions of two of these axons with swellings (large spheres with asterisk). DA axons were found to have either a large swelling along the axon (bottom reconstruction) often surrounded by medium sized swellings (green spheres) or contained terminal bulbs (top reconstruction, asterisk) reminiscent of axon retraction bulbs observed in developing neuromuscular junctions and damaged brains. Additionally, we did not observe any swellings in VTA DA dendrites despite also being sites of dopamine release (*Liu and Kaeser, 2019*) nor in any NAc dendrites or afferent axons (i.e., excitatory axons that make chemical synapses from cortical and subcortical areas) (data not shown). We next re-imaged EM

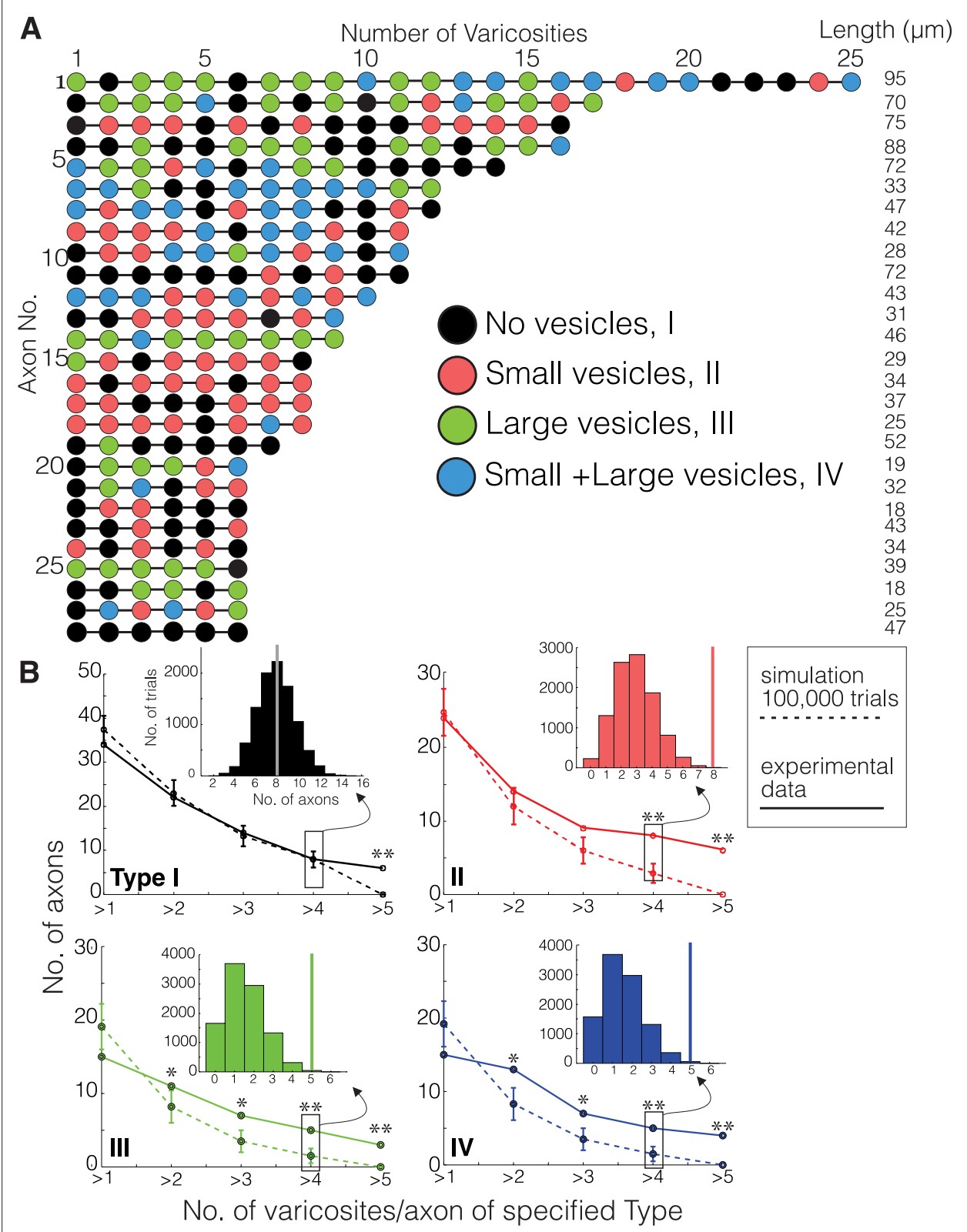

**Figure 3.** Monte Carlo simulation of dopaminergic (DA) varicosity types. (**A**) Mito-Apex2 DA axons containing six or more varicosities in the field of view are depicted with the linear order of their varicosities. Each varicosity is shown as a colored circle with each color representing a different varicosity type. The length (µm) of each reconstructed DA axon is listed on the right. (**B**) For every DA axon with three or more varicosities reconstructed from the real dataset, a simulated DA axon was created to match the number of varicosities/axon (e.g., a simulated DA axon was made with three varicosities to

*Figure 3 continued*

match a real DA axon with three varicosities). A Monte Carlo simulation was then ran 100,000 times to randomly assign varicosity types based on their population frequency reported in *Figure 2A*. For both simulated (dashed line) and real (solid line) DA axons, the number of axons was plotted against the number of varicosities/axon containing the specified type for each graph (e.g., in the top left graph, all axons containing more than one Type I varicosity were counted at the '>1' position of the x-axis). *Inset*: example of the simulated distribution and real axon (solid vertical line) data generated for these analyses for the number of axons that had more than four varicosities of each type. Asterisks denote statistically significant differences between the simulated and real data when the p-value is between 0.01 and 0.05 (*) or lower than 0.01 (**). p-Values are shown for each data point in *Figure 3—source data 1*. See *Source code 1* for MATLAB scripts used for Monte Carlo simulations.

The online version of this article includes the following source data and figure supplement(s) for figure 3:

**Source data 1.** Table of p-values for Monte Carlo simulations.

**Figure supplement 1.** Inter-varicosity distances and frequency of mitochondria at varicosity sites in dopaminergic (DA) axons.

volumes around different swellings at a higher resolution (~6 nm x, y) to better resolve the contents of these structures. The left image in *Figure 7D* shows a 2D EM image of a representative DA axon bulb (red) filled with mitochondria (two examples highlighted in blue). When this bulb was reconstructed into a 3D rendering from the serial EM sections spanning the region, we found the mitochondria to be extremely elongated, twisted, and packed into the swelling (*Figure 7D*, *right*: the bottom half of the outer membrane rendering of the bulb is removed to visualize the internal mitochondria) and that nearly all bulbs examined were similarly packed with mitochondria (data not shown).

Large axon bulbs have been most commonly associated with axonal pruning events where terminal boutons that lose their synaptic connection with a post-synaptic target form into large swellings (i.e., bulbs) and are engulfed by Schwann cells and other glial cell types presumably to remove the orphaned bouton (*Bishop et al., 2004*; *Wilton et al., 2019*). We also saw evidence of glial involvement surrounding DA axonal bulbs. *Figure 8A–B* shows a 3D reconstruction matched with a montage of EM images in *Figure 8C* of a representative DA axon bulb surrounded by numerous glial cells. Each glial cell was traced out in the volume to confirm they had characteristic non-neuronal, glial morphology (i.e., extensive branching, granules, etc., see Materials and methods) (*Fernández-Arjona et al., 2017*; *Heindl et al., 2018*; *Reichenbach et al., 2010*), and that each traced object was an individual glial cell (i.e., the reconstructed processes came from different cells). Presence of several glia around the DA axon swelling provides further evidence that these are sites of active remodeling and plasticity in response to cocaine.

Finally, the tortuous and elongated nature of the mitochondria in these bulbs made us curious about possible mitochondrial changes in other cell types in the same tissue. One advantage of large volume EM datasets is that, since all cells and intracellular organelles like mitochondria are also labeled, we could ask whether cocaine altered mitochondrial length in other parts of DA neurons as well as other neurons in the NAc. Importantly, cyto-Apex2 staining did not occlude mitochondria (see *Figure 1C*, right panel for an example) thus allowing us to measure mitochondrial lengths in cyto-Apex2 expressing DA axons.

We quantified mitochondrial lengths at five locations: in the NAc, we measured mitochondrial length in: Apex2+ DA axons, medium spiny neuron (MSN) dendrites, and chemical synapse, likely glutamatergic axons ('CS axons'), and in the VTA, Apex2+ DA soma and Apex2+ DA dendrites (*Figure 9A*, and see Materials and methods). Apex+ DA axons in the NAc had longer mitochondria throughout their arbors in the cocaine-treated mice as compared to the saline controls. *Figure 9B* shows a representative 3D reconstruction of one such DA axon with its mitochondria from the cocaine- (top and bottom left) and saline- (middle, bottom right) treated mice. When quantified across numerous axons and across mice, we find that mitochondria in DA axons from cocaine-treated mice are consistently longer than the saline controls (*Figure 9C*; mean ± SEM mitochondrial length: +saline, 0.36 ± 0.01 μm, n = 162 mitochondria across 42 axons, two mice; +cocaine, 0.79 ± 0.05 μm, n = 162 mitochondria across 35 axons, two mice. p = 7.25e-25), suggesting that Apex2 expression alone does not cause mitochondrial elongation. When we extend this analysis to other neurons in the NAc, we found that cocaine also resulted in increased mitochondrial length in MSN dendrites, the putative targets of DA axons (*Figure 9D*; mean ± SEM mitochondrial length (nm)/dendrite diameter (nm): +saline, 1.39 ± 0.12, n = 132 mitochondria across 50 dendrites, two mice; +cocaine, 3.0 ± 0.2, n = 260 mitochondria across 41 dendrites, two mice. p = 1.14e-6). However, increased mitochondrial length appeared specific to DA axons and MSN dendrites, as we observed no difference in mitochondrial length in Apex- NAc

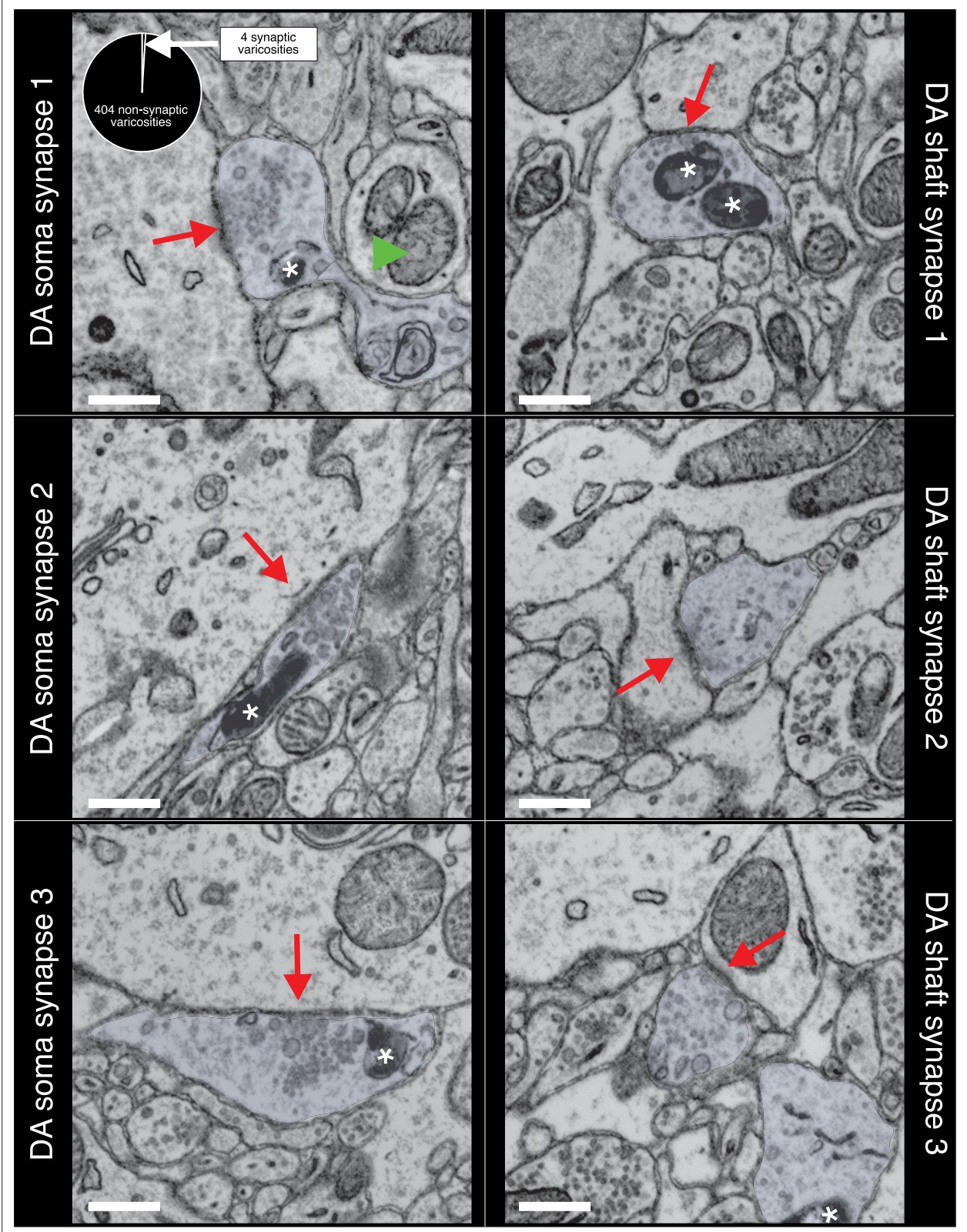

**Figure 4.** Dopaminergic (DA) axons make rare synapses on the soma and shaft of nucleus accumbens (NAc) resident neurons. Three examples of DA axons making synapses on the soma (left column) and shaft (right column) are depicted. Mito-Apex2 DA axons are shaded in light blue with an asterisk marking an Apex2-positive mitochondria which are significantly darker than Apex2-negative mitochondria (green arrowhead). Red arrows point to the post-synaptic density (PSD) formed between the DA axon and soma. Pie chart in upper left corner shows ratio of synaptic to non-synaptic DA axon

*Figure 4 continued on next page*

*Figure 4 continued*

varicosities. Scale bar = 500 nm.

The online version of this article includes the following figure supplement(s) for figure 4:

**Figure supplement 1.** Montage of identified dopaminergic (DA) axons making soma synapses.

**Figure supplement 2.** Montage of identified dopaminergic (DA) axons making shaft synapses.

**Figure supplement 3.** Montage of representative chemical synapses.

CS axons between cocaine- and saline-treated mice (*Figure 9E*; mean ± SEM mitochondrial length: +saline, 0.70 ± 0.05 µm, n = 104 mitochondria across 30 axons, two mice;+ cocaine, 0.73 ± 0.04 µm, n = 164 mitochondria across 57 axons, two mice. p = 0.64). Surprisingly, we also did not observe any differences in mitochondrial length in VTA Apex+ DA dendrites or soma as compared to controls (*Figure 9F*; mean ± SEM mitochondrial length (nm)/dendrite diameter (nm) in Apex+ DA dendrites: +saline, 1.74 ± 0.25, n = 37 mitochondria across four dendrites, one mouse; +cocaine, 1.85 ± 0.22, n = 53 mitochondria across 10 dendrites,  one mouse. p = 0.57; *Figure 9G*; mean ± SEM mitochondrial length in Apex+ DA soma: +saline, 2.46 ± 0.24 µm, n = 70 mitochondria across four soma, one mouse; +cocaine, 2.78 ± 0.23 µm, n = 141 mitochondria across five soma, one mouse. p = 0.68). Lastly, we confirmed that cocaine did not increase the density of mitochondria in DA axons (*Figure 9—figure supplement 1*; mean ± SEM # mitochondria/µm: +saline, 0.14 ± 0.02, n = 96 mitochondria across 18 axons, two mice; +cocaine, 0.16 ± 0.02, n = 107 mitochondria across 20 axons, two mice. p = 0.71). Taken together, these results indicate that cocaine sensitization increases mitochondrial length without altering mitochondrial density in the mesolimbic pathway in a cell type- (i.e., in DA axons and not in CS axons) and subcellular-specific (i.e., in DA axons and not DA soma or DA dendrites) manner.

## Discussion

We leverage the first large volume connectomic reconstruction of genetically labeled DA circuits to demonstrate that VTA DA neurons rarely make classical synapses in the medial shell of the NAc, resolving a long-standing question about whether DA communication occurs via specific synapses (*Agnati et al., 1995*; *Bérubé-Carrière et al., 2012*; *Liu et al., 2018*; *Moss and Bolam, 2008*; *Omelchenko and Sesack, 2009*; *Rice and Cragg, 2008*; *Sesack et al., 1998*). We find that most DA axons do not create classical synapses with targets in NAc and that DA axonal varicosities are primarily of four types, in descending order of frequency: (1) empty, (2) filled with small vesicles, (3) filled with large vesicles, or (4) filled with both small and large vesicles. Individual DA axons were more likely to contain stretches of a specific varicosity type than expected by chance, suggesting a previously unknown classification system for DA axons based on varicosity type. Additionally, the pleiomorphism of vesicles at individual varicosities suggests that DA axons may release different combinations of neurotransmitters at each varicosity as supported by evidence that some DA axons may release more than one neurotransmitter (*Hnasko and Edwards, 2012*; *Sulzer et al., 1998*; *Sulzer and Rayport, 2000*; *Tritsch et al., 2016*).

At many DA varicosities, we instead found evidence of complicated three-dimensional membrane invaginations with nearby axons and dendrites – 'spinules' – previously reported as an alternative method of neuronal communication and plasticity at excitatory synapses in the hippocampus and cortex (see below). After a brief exposure to cocaine, we see widespread evidence of DA axonal re-arrangements, including large-scale axonal branching and the formation of blind ended 'axonal bulbs', filled with mitochondria, and surrounded by glia. Finally, we find longer mitochondria in some cell types (e.g., DA axons but not CS axons), and in some parts of neurons but not others (e.g., DA axons but not DA dendrites or soma).

### Limitations

Our results have multiple limitations that provide important context for interpreting our results. First, our cocaine sensitization protocol is a simplified version compared to more elaborate, behavioral protocols of drug addiction. While there are clear molecular and behavioral distinctions between cocaine sensitization and experimental models of addiction (*Markou et al., 1999*; *McCutcheon et al., 2011*), we chose cocaine sensitization as the minimum experimental model to first establish a baseline

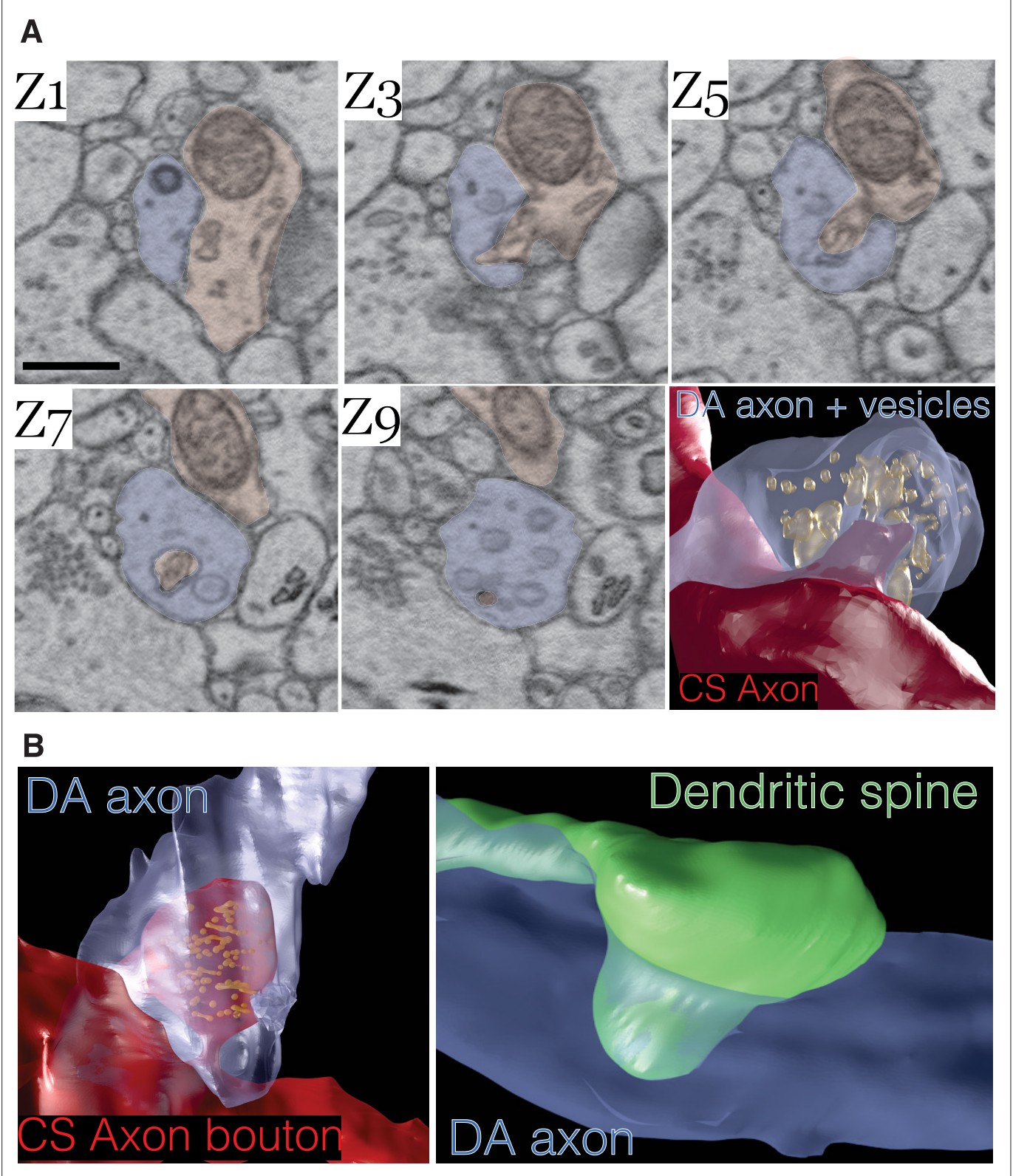

**Figure 5.** Dopamine contact points are physical interdigitations between dopaminergic (DA) axons and either afferent axons or nucleus accumbens (NAc) dendrites. (**A**) Image montage of every other electron microscopy (EM) serial section (Z1–Z9) from a mouse expressing mito-Apex2 in DA neurons. Highlighted in blue is the mito-Apex2 DA axon, and in red is the interdigitating chemical synapse (CS) axon. *Bottom right*: 3D rendering showing interdigitation of a CS axon (red) into the vesicle-filled DA axon (blue). (**B**) 3D renderings of two other examples of DA axon interdigitations. A vesicle-

*Figure 5 continued on next page*

*Figure 5 continued*

filled CS axon (red) or dendritic spine (green) interdigitate into a DA axon (blue) in the left and right images, respectively. Scale bar = (**A**) 1 µm.

The online version of this article includes the following figure supplement(s) for figure 5:

**Figure supplement 1.** Image montage of dopaminergic (DA) axon interdigitations.

for the acute effects of cocaine. Follow-up studies that examine whether these alterations persist at longer time points or in models of addiction (e.g., self-administration) would shed further light on how these re-arrangements correlate with long-term behavioral changes. Second, while we use well-documented transgenic lines that putatively targets all DA cell types indiscriminately within the VTA, we cannot rule out whether some types of axons are more efficiently labeled in our approach, (e.g., preferential expression based on AAV serotype or variability in surgical delivery of AAV). We believe these caveats are limited: we were careful to use the same batch of AAV9 virus for all experiments, we visually inspected the DAB/Apex2 reaction to ensure that the labeling intensity was approximately the same, and we also did not observe any behavioral or locomotor defects suggesting that toxicity was not an issue. Absolute labeling inefficiencies ought not to impact our characterization of dopamine axons (i.e., *Figures 1–5*) and the reproducibility of our results across animals in our cocaine studies suggests that absolute labeling inefficiencies were not so poor as to lead to confounds in statistical differences across individuals. It remains a possibility that sub-classes of DAT + neurons (e.g., different co-transmitting DA neurons, or ones that project to different regions) might have different morphological features and responses to cocaine and thus we cannot exclude the possibility that the results presented here would not be universally applicable to all DA axons. We started these investigations as a necessary first step to evaluate expression, evaluate staining, and explore how connectomic datasets could be used to detail DA circuits. Future experiments where different DA cell types are deliberately differentially labeled (e.g., mitochondria and cytoplasmic-Apex2 targeted to two DA subtypes) in the same animal would further refine these initial results. Indeed, labeling of many potential DA cell types may allow us to further refine our classification system based on the composition of varicosities in individual axons (*Figures 2–3*).

This study is also limited in the number of animals used and the number of brain regions analyzed. DA axons are widely distributed in the brain and may have fundamentally different physical interactions with target neurons in other brain regions. Future investigations using the approach highlighted here in other brain regions could help address the generalizability of these results. While limited in numbers, however, this study represents one of the largest connectomic reconstructions to date (e.g., three control and two experimental animals) (*Bates et al., 2020*; *Helmstaedter et al., 2013*; *Kasthuri et al., 2015*; *Morgan et al., 2016*; *Morgan and Lichtman, 2020*; *Motta et al., 2019*; *Vishwanathan et al., 2017*), and the differences we saw between control and cocaine exposed animals were large and consistent within cohorts across experimental conditions (i.e., no evidence of bulbs in DA axons in controls, and cocaine-treated mice show extensive DA axon branching and specific mitochondrial changes).

There are also possible confounds to our aldehyde fixation method. Aldehyde fixation, while commonly used for fixing large pieces of tissue (>300 µm), introduces fixation artifacts including distortion of cell morphology and loss of extracellular space (*Korogod et al., 2015*; *Tsang et al., 2018*) which could potentially affect our evaluation of cell-cell associations and vesicle morphology. We believe the concerns about aldehyde fixation are mitigated for several reasons. First, we find that conventional synapses (e.g., glutamatergic spine synapses) prepared in the same fashion are easily detectable with classic ultrastructural signs (i.e., large vesicle cloud, PSD) (*Figure 4—figure supplement 3*). Thus, it is unlikely we missed, for instance, dopamine varicosities forming synapses or that empty varicosities were due to artifacts of aldehyde fixation. Second, for comparisons across control and cocaine-treated mice, it is unlikely that aldehydes introduced any artifacts to our reported changes in branch number, mitochondrial length, and formation of retraction bulbs, especially given the fact that both experimental and control groups were fixed and prepared for EM on the exact same day, under the exact same conditions. Moreover, we did not make any quantitative comparisons of morphologies known to be affected by aldehyde fixation including loss of extracellular space, and proximity of glia to blood vessels and synapses (*Korogod et al., 2015*). However, we cannot rule out that aldehyde fixation somewhat confound our observations on variable vesicle morphology or the

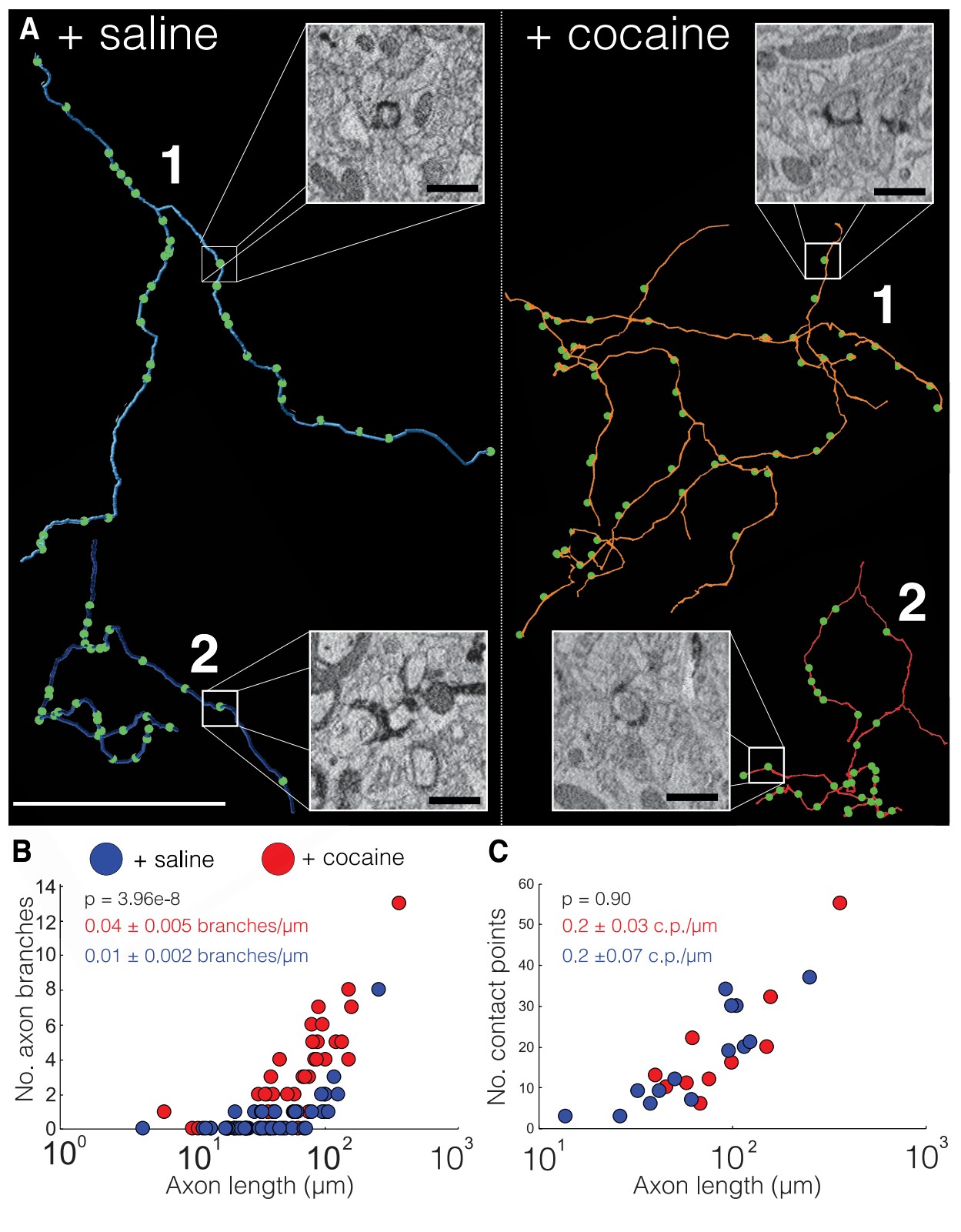

**Figure 6.** Cocaine increases branching of Apex2 dopamine axons in cocaine sensitized mice. (**A**) Two representative reconstructions of dopaminergic (DA) axons each from saline- (left, blue) and cocaine- (right, red) treated mice. Green circles represent contact points (i.e., spinules) where other neurons interdigitate with the DA axon. Electron microscopy (EM) insets: four examples of contact points identified in 20 nm resolution datasets from control and cocaine datasets. (**B**) Scatter plot of the number of DA axons branches versus axon length (μm) (+saline: 0.01 ± 0.002 branches/μm length of axon,

*Figure 6 continued on next page*

*Figure 6 continued*

n = 44 axons, two mice; +cocaine: 0.04 ± 0.005 branches/µm length of axon, n = 41 axons, two mice. p = 3.96e-8. (**C**) Scatter plot of the number of contact points (i.e., 'spinules') versus axon length (µm) + saline: 0.2 ± 0.07 contact points (c.p.)/µm length of axon, n = 240 contact points over 14 axons, two mice; +cocaine: 0.2 ± 0.03 c.p./µm length of axon, n = 142 contact points over nine axons, two mice. p = 0.90). Data: mean ± SEM. p-Values: two-tailed Mann-Whitney U test. Scale bar = (**A**) reconstructions = 40 µm, EM insets = 1 µm.

The online version of this article includes the following figure supplement(s) for figure 6:

**Figure supplement 1.** Cocaine-treated mice have increased locomotor activity.

**Figure supplement 2.** Contact points from control and cocaine 20 nm resolution datasets.

**Figure supplement 3.** Cocaine does not change the proportion of dopamine targets.

3D nature of spinules. Until recently, it was difficult to combine peroxidase staining with high-pressure freezing (HPF), a fixation method with superior ultrastructural preservation (*Korogod et al., 2015*), but recent protocols (*Korogod et al., 2015*; *Tsang et al., 2018*) suggest new experiments for leveraging HPF to more accurately reconstruct such cellular features in future experiments.

Finally, we used cyto-Apex2 for reconstructing axons and their contact points for a several reasons. First, we found that tracing axons in low-resolution EM datasets, for both controls and experimental groups, was substantially easier in cytoplasmic Apex2 axons, thus increasing our tracing throughput. In addition, we found that contact points (e.g., spinules) were also easier to detect. Detecting either a darkly Apex2 labeled cytoplasmic process in an unlabeled structure, or vice versa, was easier because of the stark contrast between Apex2 labeled and unlabeled processes. In *Figure 6* and *Figure 6—figure supplement 2* an example of this difference is shown. While it is possible that cytoplasmic Apex2 expression could potentially obscure the internal contents of varicosities (e.g., synaptic vesicles or endoplasmic reticulum), we found little evidence that cyto-Apex2 obscured the relevant ultrastructural features that were investigated here.

## The nanoscale anatomy of DA connections

How neuromodulatory neurons interact with downstream targets remains largely unknown despite their large influence on brain function (*Avery and Krichmar, 2017*; *Bargmann, 2012*; *Nadim and Bucher, 2014*). By using Apex2 labeling and large volume automated serial EM, we have conducted the first ever large volume 3D nanoscale analysis of DA axon circuitry to thoroughly investigate the anatomy of DA axons.

The existing literature characterizing DA axons has broadly relied on immunohistochemistry to label DA axons and varicosities in EM or fluorescence datasets. Previous EM studies have shown clear evidence for DA synapses on a variety of post-synaptic targets and locations, including dendritic spine heads, shafts, etc. (*Nirenberg et al., 1997*; *Pickel et al., 1981*; *Uchigashima et al., 2016*). However, many of these studies analyze smaller volumes of brains or often single EM sections, and therefore have not determined the frequency of classical ultrastructural synapses along individual DA axons; an important question when attempting to better understand DA circuitry and the relative contribution these identified synapses have to the broader circuitry of DA neurons. In addition, many of these studies were done in the rat, without genetic cell type labeling, and therefore examine boutons from DA neurons from mixed sources (e.g., substantia nigra in addition to VTA).

## Varicostiy diversity of DA connections

In our analyses, we demonstrate that 62% of all DA axon varicosities quantified contain vesicles, but that very few (<2%) made any clear synapse with any neighboring neuron. Further, varicosities varied with the types of vesicles with a majority (38%) having no vesicles, 25% with small vesicles (~48 ± 1.7 SEM nm in diameter), 19% with large vesicles (133 ± 5.5 SEM nm in diameter) and 18% with both. In all categories, DA axon varicosities contained far fewer vesicles than nearby excitatory boutons of comparable size (data not shown). Vesicles of different sizes at varicosities are sometimes indicative of release of different types of neurotransmitters (i.e., large dense-core vesicles [LCV]) are associated with neuropeptide or high molecular weight neurotransmitters and small core vesicles with classical ionotropic neurotransmitters (*Edwards, 1998*). We did not see the electron densities commonly associated with LCV, for example, in the release of catecholamines (*Grabner et al., 2005*; *Stevens et al., 2011*), suggesting that the LCV seen in DA release

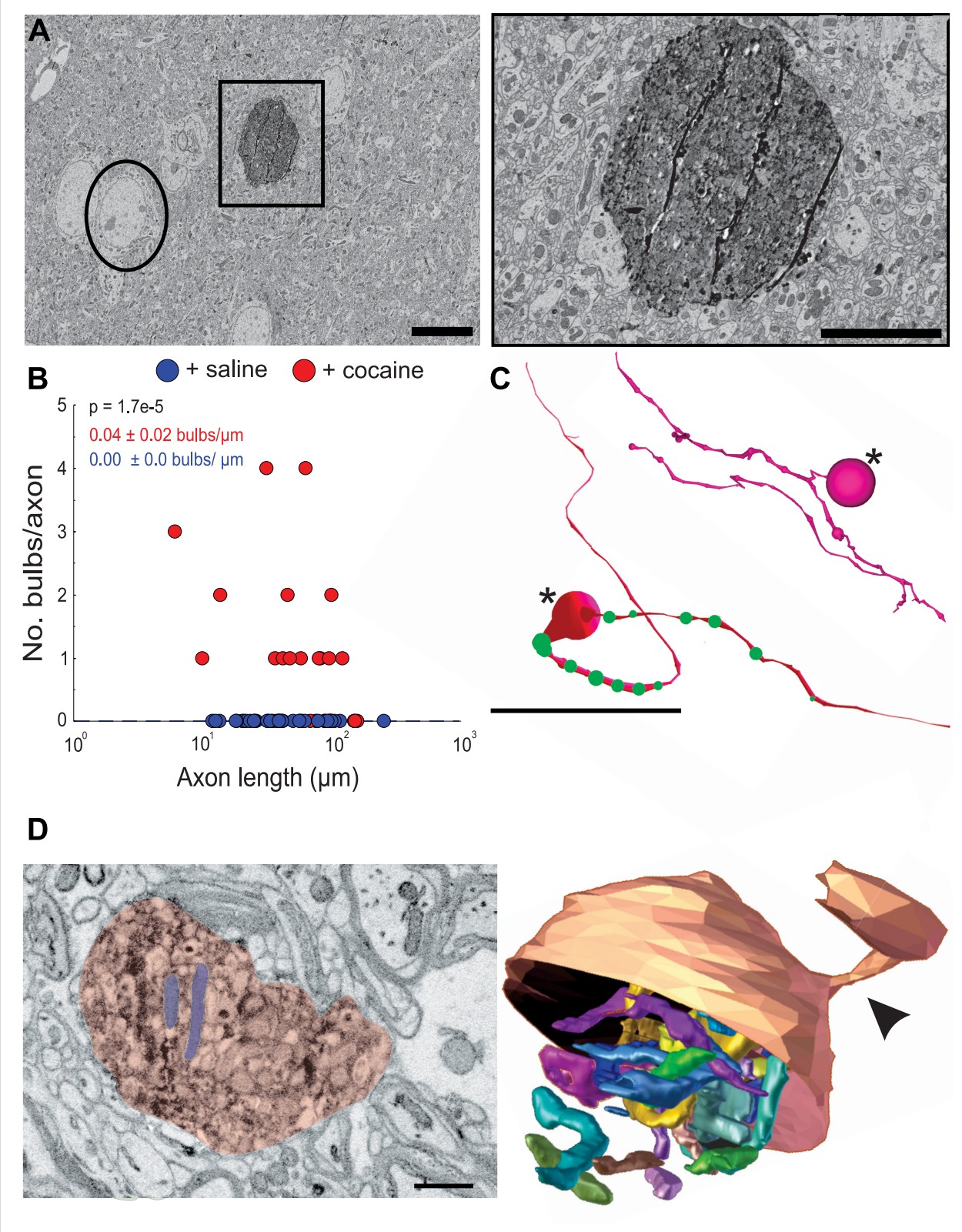

**Figure 7.** Cocaine results in the formation of large swellings filled with mitochondria in Apex2+ dopaminergic (DA) axons. (**A**) *Left*: 2D electron microscopy (EM) image of Apex2 labeled DA axon bulb in cocaine-treated mouse (square) as compared to a neighboring neuronal soma (circle). *Right*: zoomed in 2D EM image of Apex2 DA axon bulb from left panel. (**B**) Scatter plot of the number of large swellings versus axon length (μm) + saline: 0.00 ± 0.0 swellings/μm length of axon, n = 29 axons; two mice + cocaine: 0.04 ± 0.02 swellings/μm length of axon, n = 30 axons, two mice. p

*Figure 7 continued on next page*

*Figure 7 continued*

= 1.7e-5. (**C**) Reconstructions of two representative Apex+ dopamine axons with large swellings (asterisk) and medium sized swellings (green spheres). Top reconstruction depicts an axon with a terminal axon bulb and bottom reconstruction shows one along the axon. (**D**) *Left*: 2D EM image of Apex+ large DA axon swellings (red) filled with mitochondria (two examples highlighted in purple) found in the nucleus accumbens (NAc) of cocaine-treated animals. Both swellings are filled with mitochondria (examples highlighted in blue). *Right*: 3D segmentation of swelling and extremely long and coiled mitochondria found inside. Only the top half of the DA axon swelling is depicted to illustrate the mitochondria contained within. In this example, the swelling is at the end of the DA axon where it is attached to a thinner portion of the axon (arrowheads). Scale bar: (**A**) left: 10 µm, right: 5 µm, (**C**) 20 µm, (**D**) 500 nm. Data: mean ± SEM. p-Values: two-tailed Mann-Whitney U test.

might operate by different molecular release mechanisms (*Edwards, 1998*). Our results suggests that DA axons could potentially release different neurotransmitters at individual varicosities, for example, co-release of dopamine with GABA or glutamate (*Chuhma et al., 2004*; *Hnasko and Edwards, 2012*; *Sulzer et al., 1998*; *Sulzer and Rayport, 2000*; *Tritsch et al., 2016*), or release of different amounts of DA at different varicosities. One other possibility is that as this data is static, varicosities could change the composition of vesicles over time or over development. The non-random distribution of varicosity types in DA axons suggests that the release of potentially different types of transmitters is a property of individual DA axons and that DA axons might be further functionally classified by predominant varicosity type. Future experiments that, for instance, purify different vesicle sizes and identify their molecular contents could shed additional light on what these different classes functionally represent. Finally, an intriguing possibility is that nearby cells might influence the varicosity type found on individual DA axons. Such proximity analyses are likely best done in the context of knowing which cells/processes express dopamine receptors (e.g., future studies that combine molecular labeling with large volume EM; *Fulton and Briggman, 2021*; *Micheva and Smith, 2007*). This additional labeling of processes that contain DA receptors will be critical in defining the spatial extent (e.g., 'nearby') over which such analyses should be performed.

## The paucity of DA synapses

We find that only a small fraction (<2%) of all DA varicosities of any type (e.g., empty or not) made clear ultrastructural synapses with post-synaptic targets of any kind (i.e., spine, shaft, soma, etc.). The small number of DA boutons (6/410) with clear ultrastructural signs of a synapse were on the shaft and soma of resident NAc neurons, likely MSNs. While significant attention has been given to the spinous connections on MSNs, there is less data on dendritic shaft and somatic inputs onto these neurons, partly because dendritic spines are good proxies for synapses when using standard light microscopy approaches but there remain few good optical proxies for individual shaft or somatic synapses. A previous study demonstrated innervation of the shaft and soma of MSNs by boutons containing pleomorphic synaptic vesicles (*Wilson and Groves, 1980*), and we extend that work demonstrating that some of that innervation is likely from DA axons. Finally, a current theory of DA transmission is that while DA axons make specific ultrastructural synapses, there remains a 'mismatch' between presynaptic elements and post-synaptic dopamine receptors (*Agnati et al., 1995*), therefore invoking the idea of volume transmission. Our data clearly suggest differences with that model: the absence of ultrastructural evidence of DA synapses writ large. We define here 'synapse' in the classical anatomical sense with ultrastructural evidence of presynaptic vesicles and post-synaptic specializations (e.g., parallel membranes or a PSD). This definition leverages decades of combined EM and light microscopic work correlating EM signatures with molecular machinery (e.g., active zones contain presynaptic $Ca^{2+}$ channels and PSDs contains synaptic receptors). This work concludes primarily that most DA boutons show little evidence of classic EM signatures of synapses. It remains possible that molecular release machinery exists at these boutons without ultrastructural synaptic specialization, perhaps a fundamentally different type of NT release than classic inotropic transmission. Indeed, there is some evidence that DA axons contain release machinery, and, interestingly, that this machinery (i.e., RIM, ELK5) is only found in ~30% of dopamine varicosities (*Liu et al., 2018*) further highlighting the complex nature to dopamine release. Thus, our results are a steppingstone to future correlated molecular and EM studies leveraging a range of new protocols (*Fulton and Briggman, 2021*; *Micheva and Smith, 2007*)

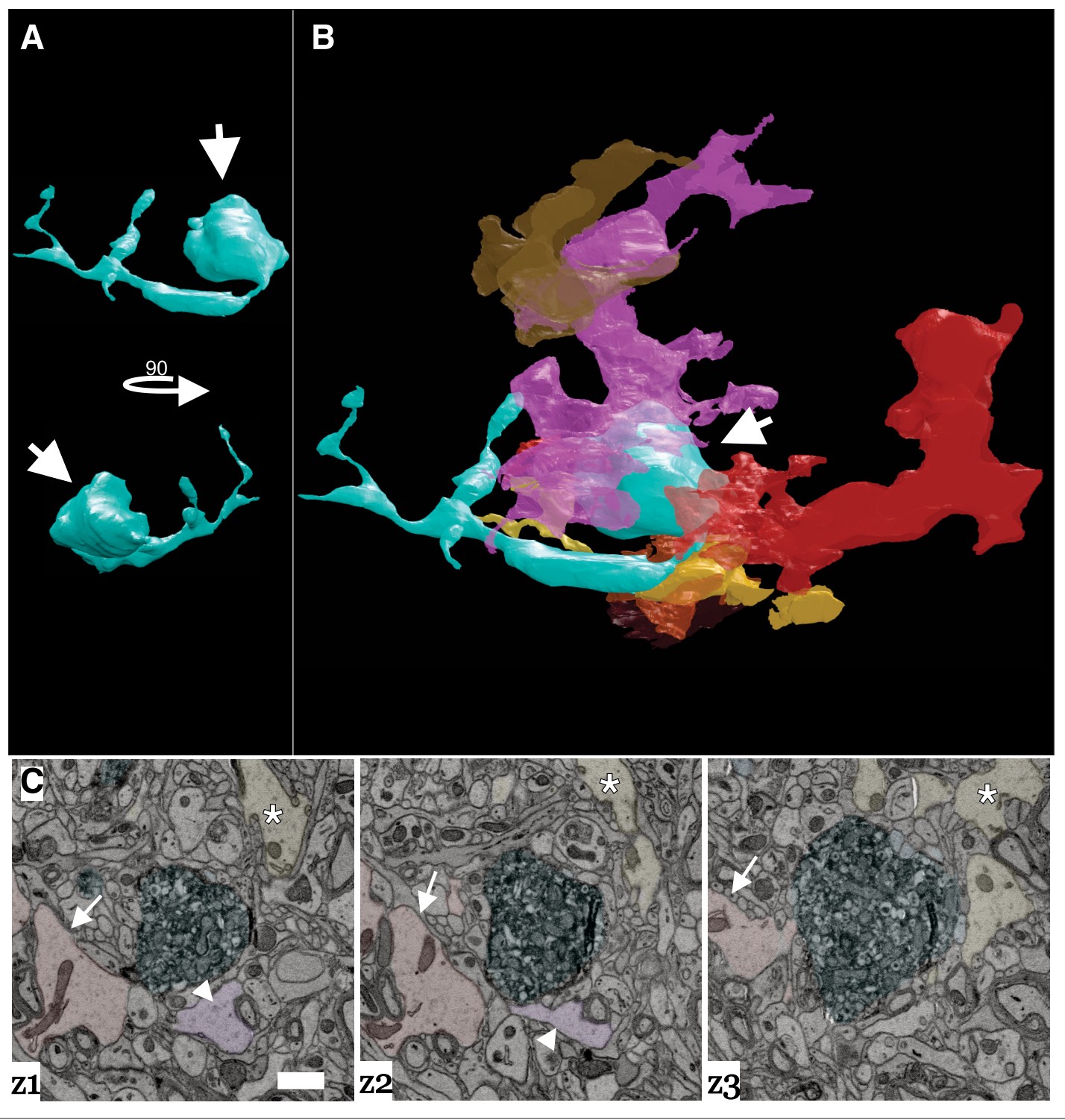

**Figure 8.** Dopamine axon swellings are surrounded by glia. (**A**) 3D reconstruction of a dopamine axon swelling from a cocaine sensitized mouse. Bottom image is rotated 90 degrees relative to top view. White arrows point to swelling. (**B**) 3D reconstruction of dopamine axon swelling (arrow) from (**A**) with glia surrounding it. Each differently colored object represents a different glial cell. (**C**) Montage of three serial electron microscopy (EM) images color-coded to highlight the dopamine axon swelling (green), and three example glial cells (red/arrow, pink/arrowhead, yellow/asterisk) that correspond to (**A**) and (**B**). Scale bar: (**C**) 1 μm.

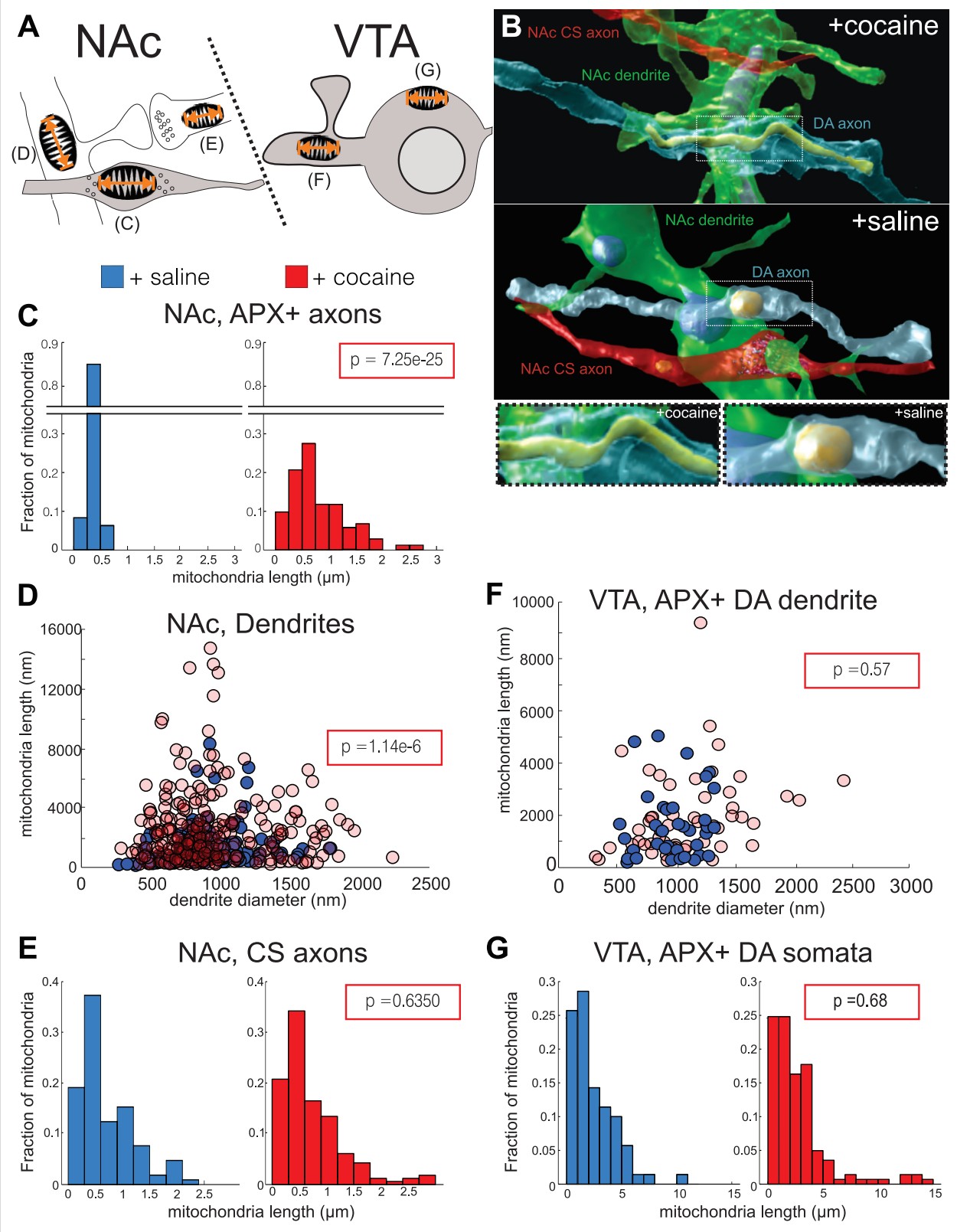

**Figure 9.** Cocaine results in increased mitochondrial length in dopaminergic (DA) axons and medium spiny neuron (MSN) dendrites. (**A**) Cartoon depicting neurons where mitochondrial length was measured. *Left:* in the nucleus accumbens (NAc), we measured mitochondria in: (**C**) Apex2+ DA axons, (**D**) MSN dendrites, and (**E**) excitatory axons making chemical synapses (chemical synapse [CS] axons). *Right*: in the ventral tegmental area (VTA), we measured mitochondria in: (**F**) Apex2+ DA dendrites, and (**G**) Apex2+ DA soma. (**B**) 3D reconstructions from cocaine- (top) and saline- (bottom)

*Figure 9 continued on next page*

*Figure 9 continued*

treated mice of a DA axon (blue), afferent CS axon (red), and MSN dendrite (green) from the NAc. Shown is just the mitochondria reconstructed within the DA axon. Bottom images show zoomed views centered on the DA axon mitochondria from cocaine- (left) and saline- (right) treated mice. (**C**) Histograms of mitochondrial length in NAc, Apex2+ ('APX+') DA axons +saline: 0.36 ± 0.01 µm, n = 162 mitochondria across 42 axons, 2 mice; +cocaine: 0.79 ± 0.05 µm, n = 162 mitochondria across 35 axons, two mice. p = 7.25e-25. (**D**) Scatter plot of mitochondrial lengths versus dendrite diameter of NAc MSN dendrites (+saline: 1.39 ± 0.12 mitochondrial length (nm)/dendrite diameter (nm), n = 132 mitochondria across 50 dendrites, two mice; +cocaine: 3.0 ± 0.2 µm mitochondrial length (nm)/dendrite diameter (nm), n = 260 mitochondria across 41 dendrites, two mice. p = 1.14e-6). (**E**) Histogram of mitochondrial lengths in NAc CS axons (+saline: 0.70 ± 0.05 µm, n = 104 mitochondria across 30 axons, two mice; +cocaine: 0.73 ± 0.04 µm, n = 164 mitochondria across 57 axons, two mice. p = 0.64). (F) Scatter plot of mitochondrial lengths versus dendrite diameter of VTA Apex2+ DA dendrites + saline: 1.74 ± 0.25 mitochondrial length (nm)/dendrite diameter (nm), n = 37 mitochondria across four dendrites, one mouse; +cocaine: 1.85 ± 0.22 mitochondrial length (nm)/dendrite diameter (nm), n = 53 mitochondria across 10 dendrites, one mouse. p = 0.57. (**G**) Histogram of mitochondrial length in Apex2+ DA Soma (+saline: 2.46 ± 0.24 µm, n = 70 mitochondria across four soma, one mouse; +cocaine: 2.78 ± 0.23 µm, n = 141 mitochondria across five soma, one mouse. p = 0.68). Blue histograms and data points on scatter plots = +saline-treated mice, and red histograms and data points = +cocaine-treated mice. Data: mean ± SEM. p-Value: two-tailed Mann-Whitney U test.

The online version of this article includes the following figure supplement(s) for figure 9:

**Figure supplement 1.** Cocaine does not change the number of mitochondria in dopaminergic (DA) axons.

that investigate how the presence of molecular machinery correlate with EM ultrastructure including different varicosity types we describe.

## Spinules as a possible mechanism of DA transmission

The other major type of direct physical interactions we found were numerous instances of axon and dendrites physically interdigitating with the membranes of DA varicosities. These interdigitations most closely resembled previously characterized synaptic structures termed 'spinules' which occur in cortical and hippocampal neurons, and in both in vitro cell cultures and brain slices (*Petralia et al., 2015*; *Spacek and Harris, 2004*; *Zaccard et al., 2020*). Spinules are small (~100 nm diameter) membranous protrusions typically found at or near synaptic sites at dendritic spine synapses. Spinules most often emerge from dendritic spines and to a lesser extent axonal boutons, but most often these spinules, regardless of their source, project into axons – either the presynaptic axon or a neighboring axon not actively forming a synapse (*Spacek and Harris, 2004*). Furthermore, spinules have been detected using HPF methods, free of any aldehydes, and in dissociated cultures of living neurons, suggesting that these invaginations are not artifacts caused by chemical fixation methods (*Tao-Cheng et al., 2009*; *Zaccard et al., 2020*). While their function is not totally known, spinule formation seems activity dependent (e.g., spinules are induced by K+-mediated neuronal depolarization, NMDA [*Tao-Cheng et al., 2009*], and glutamate activity *Richards et al., 2005*). Our results now show, for the first time, DA neurons also form spinules. However, unlike hippocampal neurons where spinules more often originate from dendrites, we find that 83% of spinules involving DA axons originate from unlabeled axons, presumably afferent axons from cortex or thalamus. Finally, we found that cocaine exposure does not change the frequency of spinules along DA axons (i.e., spinules/µm), but as DA axons branch more in response to cocaine, the newly formed axons also contain more spinules and thus more physical interactions with surrounding neurons. Labeling of other neuromodulatory pathways (e.g., cholinergic, serotonergic, etc.) could reveal whether such spinule-like structures are a common structural motif at neuromodulatory varicosities.

## Cocaine-induced DA circuit remodeling

### Large-scale DA axon remodeling

We find evidence of large-scale anatomical changes in DA axons following exposure to cocaine. Broadly, our data are consistent with active remodeling of dopamine axons: retraction bulbs on axon terminals in the process of pruning (i.e., removing connections) and increased axonal branching (i.e., formation of new connections). Additionally, coinciding with these structural plasticity events, we find evidence of alterations in mitochondria, potentially due to changes in cellular metabolism. While previous reports have focused on changes in DA spine density in the VTA and MSN dendritic spine synapse density in the NAc (*Alcantara et al., 2011*; *Barrientos et al., 2018*; *Li et al., 2004*; *Sarti*

*et al., 2007*), our results fill in an important gap to our understanding of how cocaine alters the structure of DA axons. Notably, these changes occurred after just 4 days of exposure and 4 days of withdrawal, suggesting a rapid pace of remodeling.

The most striking change in response to cocaine we discovered were large swellings in DA axons. These swellings bare striking similarities to remodeling events observed during development and traumatic brain injury (TBI). Axonal bulbs in development and TBI are distinguished from each other as either axon retraction or axon degeneration (i.e., 'Wallerian degeneration'), respectively (*Bishop et al., 2004*; *Johnson et al., 2013*; *Rosenthal and Taraskevich, 1977*). We do not observe axon fragmentation nor removal of the entire axon, both hallmarks of axon degeneration, but rather the bulbs we observe are connected to otherwise intact DA axons and thus resemble more closely axonal pruning seen during development. However, unlike developmental axonal arbor plasticity, which occurs generally as a monotonic decrease in the size of axonal arbors over development (*Hubel et al., 1977*; *LeVay et al., 1980*; *Tapia et al., 2012*), we see signs of both increased branching and possible retraction bulbs suggesting that different cellular mechanisms could mediate the plasticity described here. It would be interesting to see if other types of circuit plasticity follow similar principles.

These results raise several new questions: (1) what drives large bulb formation, (2) how are DA bulbs different from glutamatergic axon bulbs that are thought to form pruned synapses, given that DA axons rarely make conventional synapses, and (3) what, if any, release of neurotransmitter occurs at bulbs?

## Glia and DA axonal remodeling

The presence of 'activated' glia around the DA axon bulbs further suggests these are areas of the DA axon being actively sensed and responded to by the brain's glial monitoring system. While it is established that cocaine and other drugs of addiction can activate glial cells (*Linker et al., 2019*; *Miguel-Hidalgo, 2009*), the direct role for glia in mediating the effects of drug abuse has remained unclear. Here, for the first time, we provide direct anatomical evidence for a potential role for glia following cocaine exposure in mediating retraction bulb development and resolution, like how glial cells mediate axonal retraction bulbs during development (*Bishop et al., 2004*; *Wilton et al., 2019*). Because our sample size is limited to qualitative observations, future experiments that manipulate the presence or activity of glia (*De Luca et al., 2020*; *Nguyen et al., 2020*; *Paolicelli et al., 2017*) will provide potentially causal roles glia have in DA axons remodeling after cocaine exposure.

## Cell type and cell compartment-specific mitochondrial remodeling after cocaine exposure

In addition to observing signs of plasticity in DA axons, we also find evidence that cocaine can result in alterations at the subcellular level through changes in mitochondria. Previous studies have demonstrated that cocaine alters brain energy homeostasis, including changes in oxidative stress, cellular respiration, and enrichments of mitochondrial-related transcripts in NAc brain slice preps (*Dietrich et al., 2005*; *Feng et al., 2014*; *Kalivas, 2009*; *Lehrmann et al., 2003*; *Volkow et al., 1991*). The changes in mitochondrial length documented here could provide a potential cellular substrate for these energy changes (*Glancy et al., 2020*; *Skulachev, 2001*). Another possibility is that mitochondrial elongation is the result of hyperactivity of the mesolimbic circuit, as demonstrated by the increased movement of mice exposed to cocaine (*Figure 6—figure supplement 1*). Changes in mitochondrial cytochrome oxidase staining with increased neuronal activity are well known (i.e., cytochrome oxidase and ocular dominance columns in cortex) (*Horton and Hedley-Whyte, 1984*; *Wong-Riley and Riley, 1983*) and, if so, the cell type-specific nature of our results (i.e., DA axons and MSN dendrites show elongated mitochondria but excitatory, putative glutamate, axons and DA soma and dendrites of the VTA do not) suggest that cocaine may act preferentially on DA axons and MSNs, at least with regard to cellular metabolism. Finally, since many of these results could not have been collected in any other way, we conclude that large volume connectomic efforts with cell type specific labeling could be a valuable tool for the study of other neuromodulatory circuits.

# Materials and methods

**Key resources table**

| Reagent type (species) or resource | Designation | Source or reference | Identifiers | Additional information |
|---|---|---|---|---|
| Genetic reagent (*Mus musculus*) | *Slc6a3<sup>tm1(cre)Xz</sup>*/J | Provided by the Xiaoxi Zhuang lab (The University of Chicago), PMID:15763133 | *RRID:IMSR_ JAX:020080* | Also available at Jackson Laboratories: https://www.jax.org/ strain/020080 |
| Chemical compound, drug | 3,3'-Diaminobenzidine | Sigma-Aldrich | D12384 | 50 mg/ml |
| Chemical compound, drug | Sodium cacodylate buffer, pH 7.4 | Electron Microscopy Sciences | 11653 | 0.2 M stock, use at 0.1 M |
| Chemical compound, drug | Sodium hydrosulfite | Sigma-Aldrich | 157953 | 0.8% (w/v) |
| Chemical compound, drug | Sodium bicarbonate | Sigma-Aldrich | S5761 | 0.1 M stock, used at 60% (v/v) |
| Chemical compound, drug | Sodium carbonate | Sigma-Aldrich | S7795 | 0.1 M stock, used at 40% (v/v) |
| Chemical compound, drug | Osimum tetroxide | Electron Microscopy Sciences | 19150 | 4% aqueous stock solution, use at 2% |
| Chemical compound, drug | Paraformaldehyde | Electron Microscopy Sciences | 15710 | 16% aqueous stock solution, use at 2% |
| Chemical compound, drug | Glutaraldehyde | Electron Microscopy Sciences | 16220 | 25% aqueous stock solution, use at 2.5% |
| Chemical compound, drug | Postassium ferrocyanide | Sigma-Aldrich | P3289 | 2.5% |
| Chemical compound, drug | Pyrogallol | Sigma-Aldrich | P0381 | 4% |
| Chemical compound, drug | Uranyl acetate | Electron Microscopy Sciences | 22400–4 | 4% aqueous stock solution, use at 1% |
| Chemical compound, drug | Lead (II) nitrate | Sigma-Aldrich | 228621 | 0.66% |
| Chemical compound, drug | Embed 812 kit | Electron Microscopy Sciences | 14120 | 49% Embed 812, 28% DDSA, 21% NMA, and 2.0% DMP 30 |
| Chemical compound, drug | Cocaine | Obtained through DEA license (Xiaoxi Zhuang, The University of Chicago) | | 10 mg/kg |

## Animals and AAV

DAT-CRE mice ~15 weeks used in this study were acquired from Xiaoxi Zhuang (The University of Chicago) and can also be found at Jackson Laboratory (https://www.jax.org/strain/020080). The mouse line used targets Cre recombinase to the *Slc6a3* locus that encodes DAT to create the DAT-Cre knock-in strain. As a result, it disrupts one copy of DAT. AAV-CAG-DIO-APEX2NES (Cyto-Apex) was acquired as a gift from the laboratory of Joshua Sanes (Harvard) and is now available at Addgene: #79907. AAV-CAG-DIO-APEX2-MITO was generated in our lab by cloning the mitochondrial targeting sequencing from mito-V5-APEX2 (Addgene #72480) and placing it on the 5' end of APEX2-NES in AAV-CAG-DIO-APEX2NES. Finally, the nuclear export sequence (NES) was removed from APEX2. AAV9 virus was generated at the University of North Carolina School of Medicine Vector Core facility (https://www.med.unc.edu/genetherapy/vectorcore/). Animal care, perfusion procedures, and AAV injections were followed according to animal regulations at the University of Chicago's Animal Resources Center (ARC) and approved IACUC protocols. The following IACUC numbers are associated with the listed experimentation performed. ACUP #72480: animal perfusion and EM preparation of brain tissue. DEA license: RZ0294102 was issued to Xiaoxi Zhuang was used to obtain and administer cocaine. ACUP #72194 and IBC #1181: cocaine administration, AAV delivery, and locomotor tracking experiments.

## AAV injections

AAV9 injections were performed using a standard stereotaxic frame; 70-100 nl of virus (~2.9×10$^{12}$ viral genomes/ml) were bilaterally injected into the VTA using the stereotactic coordinates: 3.1 posterior of bregma, 0.55 lateral bregma, and 4.4 ventral of the dura. Mice were aged 4 weeks, based on prior experience with APEX expression (*Sampathkumar et al., 2021a*) to allow for AAV expression before perfusion or cocaine sensitization experiments.

## Cocaine sensitization

Mice were given a once daily IP injection of either cocaine (10 mg/kg) or equivalent volume of saline every other day for a total of four injections. Immediately following each injection, mice were placed in a novel environment where their locomotor activity was automatically monitored for 1 hr and then returned to their home cage.

## Apex2 staining and EM preparation

Brains were prepared in the same manner and as previously described (*Hua et al., 2015*). Briefly, an anesthetized animal was first transcardially perfused with 10 ml 0.1 M sodium cacodylate (cacodylate) buffer, pH 7.4 (Electron Microscopy Sciences [EMS]) followed by 20 ml of fixative containing 2% para-formaldehyde (EMS), 2.5% glutaraldehyde (EMS) in 0.1 M sodium cacodylate (cacodylate) buffer, pH 7.4 (EMS). The brain was removed and placed in fixative for at least 24 hr at 4°C. To ensure that the same brain region was isolated across replicates, the brain was mounted to the vibratome on the anterior side and carefully trimmed down through the cerebellum until it just touched the posterior cortex. The brain was then completely sectioned into a series of 300 µm vibratome sections, placed, in order, into a 24-well dish, and incubated into fixative for 24 hr at 4°C. Apex2 precipitation and polymerization was performed by washing slices extensively in cacodylate buffer at room temperature, incubated in 50 mg/ml DAB for 1 hr at room temperature followed with DAB/0.03% (v/v) H$_2$0$_2$ until a visible precipitate forms (15–20 min). Slices were washed extensively in cacodylate buffer. Slices were reduced in 0.8 %(w/v) sodium hydrosulfite in 60% (v/v) 0.1 M sodium bicarbonate 40% (v/v) 0.1 M sodium carbonate buffer for 20 min and washed in cacodylate buffer. To ensure the same region of the NAc was taken across animal samples, we used areal landmarks of the anterior commissure, shape of the corpus callosum, and boundary of positive and negative Apex2 staining in the medial portion of the brain to isolate the medial shell of the NAc. The medial shell of the NAc and a piece of tissue spanning the entire VTA were excised and prepared for EM by staining sequentially with 2% osmium tetroxide (EMS) in cacodylate buffer, 2.5% potassium ferrocyanide (Sigma-Aldrich), 4% pyrogallol, unbuffered 2% osmium tetroxide, 1% uranyl acetate, and 0.66% aspartic acid buffered lead(II) nitrate with extensive rinses between each step with the exception of potassium ferrocyanide. The sections were then dehydrated in ethanol and propylene oxide and infiltrated with 812 Epon resin (EMS, mixture: 49% Embed 812, 28% DDSA, 21% NMA, and 2.0% DMP 30). The resin-infiltrated tissue was cured at 60°C for 3 days. Using a commercial ultramicrotome (Powertome, RMC), the cured block was trimmed to an ~1.0 mm × 1.5 mm rectangle and ~2000, 40-nm-thick sections were collected from each block on polyamide tape (Kapton) using an automated tape collecting device (ATUM, RMC) and assembled on silicon wafers as previously described (*Kasthuri et al., 2015*). The serial sections were acquired using backscattered electron detection with a Gemini 300 scanning electron microscope (Carl Zeiss), equipped with ATLAS software for automated wafer imaging. To further ensure we captured the same region of the NAc across datasets, we used the anterior commissure, which was contained within our serial sections, as a guide for were to reproducibly place the ROI during imaging. Dwell times for all datasets were 1.0 µs. For 20 and 6 nm resolution datasets, sections were brightness/contrast normalized and rigidly aligned using TrakEM2 (FIJI) *Cardona et al., 2012* followed by non-linear affine alignment using AlignTK (https://mmbios.pitt.edu/aligntk-home) on Argonne National Laboratory's super computer, Cooley. Different image processing tools were packaged into Python scripts that can be found here: https://github.com/Hanyu-Li/klab_utils (*Li, 2021*). Final volumes collected for each mouse were: (1) Mito-Apex = 160 µm × 160 µm × 11 µm @ 6 nm, 1 mouse, (2) Cyto-Apex = 450 µm × 350 µm × 30 µm @ 20 nm, 4 mice and 130 µm × 130 µm × 10 µm @ 6 nm, 4 mice.

## Data analysis

Aligned datasets were manually skeletonized and annotated using the publicly available software, Knossos (https://knossos.app). All quantifications reported are mean ± standard error of the mean (SEM) and reported across results collected from an EM volume dataset collected and annotated from each animal (i.e., two biological replicates for control, two biological replicates for cocaine-treated mice). Two-tailed Mann Whitney U statistics test was used to test for significance (*Marx et al., 2016*). Data annotations were done by two individuals (GAW, AMS) who were not knowledgeable of whether they were annotating data from the control or cocaine datasets. To ensure accuracy of the data, ~33% of the annotations from one person was verified by the other without knowledge of whether the data came from the control or experimental group. We found a >98% agreement between manual annotators. Because of the rarity of putative DA synapses, we had a third expert EM annotator re-examine 25 randomly chosen annotated boutons along mito-Apex2 labeled DA axons and found 24/25 of the boutons made no signs of classical synapses. Classes of cell types were identified by distinguishing anatomical properties: Apex2+ DA neurons (soma and dendrites in the VTA and axons in NAc) were identified by their dark precipitate, medium spiny neurons in the NAc by the presence of numerous dendritic spines, and chemical synapse axons in the NAc by their formation of synapses on dendritic spines. We defined boutons or varicosities, here used interchangeably, as large abrupt swellings along the length of axons (i.e., the diameter of the axon increases ~2-fold or more in ~1 µm of distance or less), consistent with previous classifications (*Bodian, 1970*). We classified whether varicosities were synaptic if we found ultrastructural evidence for synaptic vesicles contained in the varicosity, if the vesicles are clustered, clear sign of a PSD, parallel pre- and post-synaptic membranes, and presynaptic active zones. We classified varicosities as synaptic if we could find evidence for two or more of the above, preferably one presynaptic measure and one post (e.g., synaptic vesicles clustered opposed to a strong PSD), in any object with membrane to membrane apposition anywhere along the defined varicosity. Skeleton information was exported into ma separated matrices using homemade Python scripts that compute skeleton features from the Knossos xml annotation file. Code is freely available here: https://github.com/knorwood0/MNRVA, (*Norwood, 2020*, copy archived at swh:1:rev:d-c5512ff7c7ce6d5cd1de5bd7a8678193cdcf750). Quantification and plotting of different anatomical features were performed in MATLAB and excel.

## DA varicosity vesicle diameter

Vesicle diameter was measured in Knossos by scrolling through the image stack to find the center point of the vesicle (i.e., image where diameter was the largest). A node was then placed over the vesicle and sized to match the vesicle size. Node radius was then used to calculate diamgeter in pixels and then converted to nanometers. To ensure our method was accurate, 25% of the vesicles measured in Knossos were also measured in Fiji/ImageJ using the Set scale (set at 6 nm/pixel) and measured using the line tool.

## DA axon varicosities

DA axon varicosities were identified as regions where the DA axon diameter increased ~3-fold (~100 to ~300 nm in axon diameter). Small vesicles were scored as those ranging in diameter of 26–67 nm and large vesicles ranged in diameter of 90–238 nm.

## DA axon spinules

Spinules in the mito-Apex2 datasets were identified as regions of membrane invaginations into DA axons that could be traced out to a nearby neurite. Spinules in cyto-Apex2 datasets were identified as invaginations that were void of Apex2 staining that invaginated into the Apex2+ DA axon and could be traced out to a nearby neurite. Spinule invaginations were further confirmed by observing a neurite entering a DA axon and disappearing in the image stack using the three orthogonal views of Knossos. Neurites that passed over DA axons were not scored as contact points/spinules.

## Chemical synapse identification

In the mito-Apex2 6 nm resolution dataset, 100 boutons containing a prominent vesicle cloud containing numerous (~>50) small vesicles were marked in Knossos as putative excitatory glutamatergic axonal boutons. Boutons were then visually inspected in z for the presence of a PSD.

## Monte Carlo simulations of DA axon varicosity types

Monte Carlo simulations were ran using custom scripts in MATLAB and are included in this publication as a source code (*Source code 1*).

## Inter-varicosity distance

Using Knossos, every varicosity was annotated as empty, small vesicle, large vesicle, or small+large vesicle within a given reconstructed DA axon. DA axon reconstructions were made using skeletons that accurately followed the tortuosity of the axon. Using the axonal reconstructions and varicosity annotations, the Euclidean distance along the axonal reconstruction between each varicosity pair was calculated in nanometer units using homemade scripts (https://github.com/knorwood0/MNRVA; *Norwood, 2020*).

## Mitochondrial scoring

Using Knossos, every annotated varicosity was marked for the presence or absence of a mitochondria contained within ~1 μm of the varicosity.

## DA axon branching

Cyto-Apex2 DA axons were manually reconstructed and the number of branches each axon made was scored.

## Large DA axon swellings

The entire volume of each dataset (~0.5 × 0.5 × 0.0 2 mm) was surveyed for the presence of retraction bulbs. Additionally, the entire EM section (~1 × 1.5 mm) was spot-checked by manually looking across every 20th section.

## Glia identification

Glia were identified by several criteria: extensive branching, cytoplasmic granules, vacuoles, and lack of any synapses.

## Mitochondrial length

The length of mitochondria was measured in Knossos by placing nodes along the mitochondria's longest axis from one of the three orthogonal views (xy,xz,yx). Homemade scripts (https://github.com/knorwood0/MNRVA; *Norwood, 2020*) were then used to measure the distance between nodes. Dendrite diameter was measured by centering the mitochondria in the Knossos viewing window and then placing node pairs across the width of the dendrite in all three orthogonal axes. The average between the three measured lengths was the value used to report dendrite diameter.

## 3D rendering

3D manual segmentation was done using VAST (*Berger et al., 2018*) and exported using MATLAB as.OBJ files which were then imported and rendered using Blender 2.79.

## Additional information

### Funding

| Funder | Grant reference number | Author |
| --- | --- | --- |
| McKnight Foundation | | Gregg Wildenberg<br>Anastasia Sorokina<br>Bobby Kasthuri |
| National Institutes of Health | U01 MH109100 | Gregg Wildenberg<br>Anastasia Sorokina<br>Bobby Kasthuri |

| Funder | Grant reference number | Author |
|---|---|---|
| National Science Foundation | | Gregg Wildenberg<br>Anastasia Sorokina<br>Bobby Kasthuri |

The funders had no role in study design, data collection and interpretation, or the decision to submit the work for publication.

## Author contributions

Gregg Wildenberg, Conceptualization, Data curation, Formal analysis, Investigation, Methodology, Project administration, Resources, Software, Supervision, Validation, Visualization, Writing - original draft, Writing - review and editing; Anastasia Sorokina, Formal analysis; Jessica Koranda, Xiaoxi Zhuang, Daniel McGehee, Methodology; Alexis Monical, Chad Heer, Investigation; Mark Sheffield, Resources, facilitated access to experimental resources involving AAV expression in mice; Bobby Kasthuri, Conceptualization, Funding acquisition, Supervision, Validation, Writing - original draft, Writing - review and editing

## Author ORCIDs

Gregg Wildenberg (iD) http://orcid.org/0000-0003-2550-5497
Bobby Kasthuri (iD) http://orcid.org/0000-0003-3825-931X

## Ethics

Animal care, perfusion procedures, and AAV injections were followed according to animal regulations at the University of Chicago's Animal Resources Center (ARC) and approved IACUC protocols. The following IACUC numbers are associated with the listed experimentation performed. ACUP #72480: animal perfusion, and electron microscopy preparation of brain tissue. DEA license: RZ0294102 was issued to Xiaoxi Zhuang was used to obtain and administer cocaine. ACUP #72194 and IBC #1181: cocaine administration, AAV delivery, and locomotor tracking experiments.

## Decision letter and Author response

Decision letter https://doi.org/10.7554/eLife.71981.sa1
Author response https://doi.org/10.7554/eLife.71981.sa2

# Additional files

## Supplementary files

- Transparent reporting form
- Source code 1. MATLAB code to generate *Figure 3B* Monte Carlo simulations.

## Data availability

Homemade code used for EM image processing (i.e., 2D stitching, 3D alignments, brightness/contrast normalization) can be freely accessed here: https://github.com/Hanyu-Li/klab_utils. 3D alignment was performed using the publicly available Aligntk software available here: https://mmbios.pitt.edu/align-tk-home. Homemade code used for data anlaysis of Knossos traced skeletons (i.e., xml files) can be freely accessed here: https://github.com/knorwood0/MNRVA (copy archived at swh:1:rev:dc5512ff7c-7ce6d5cd1de5bd7a8678193cdcf750). All EM data is are freely available online and can be accessed here: https://bossdb.org/project/wildenberg2021. Datasets are from the following brain regions and contains the in-plane imaging resolution listed in the filename.

The following dataset was generated:

| Author(s) | Year | Dataset title | Dataset URL | Database and Identifier |
|---|---|---|---|---|
| Wildenberg G, Sorokina A, Koranda J, Monical A, Heer C, Sheffield M, Zhuang X, McGehee D, Kasthuri B | 2021 | Combined genetic labeling and connectomics of dopamine axons to reveal wiring properties and rewiring affects caused by cocaine | https://bossdb.org/project/wildenberg2021 | BossDB, wildenberg2021 |

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
