## [Editor Report]

How dopaminergic neurons communicate with other neurons, whether via point-to-point contact involving classical synapses or through volume transmission, has remained controversial. By performing large-scale serial electron microscopy combined with genetic labelling methods, this study reveals that dopaminergic axonal varicosities lack features of classical synapses and that following exposure to cocaine dopaminergic axons undergo extensive remodeling. These findings provide major insights into the biology of dopaminergic axons and are of fundamental interest, and they also form a basis for understanding dopaminergic circuit changes associated with drugs of abuse.

---

## [Decision Letter]

**Decision letter after peer review:**

Thank you for submitting your article "Cell type specific labeling and partial connectomes of dopaminergic circuits reveal non-synaptic communication and large-scale axonal remodeling after exposure to cocaine" for consideration by *eLife*. Your article has been reviewed by 3 peer reviewers, including Yukiko Goda as Reviewing Editor and Reviewer #3, and the evaluation has been overseen by Michael Taffe as the Senior Editor.

Essential revisions:

All three reviewers acknowledge the valuable nature of the present EM data of DA axons. However, two major concerns have been raised regarding the interpretation of axonal varicosities.

(1) An unequivocal definition for the classification of synaptic vs. non-synaptic varicosities is needed. Also, in order to exclude the possibility that the sparsity of classical synapses is due to some imaging artefact, include reconstructions of non-DA axons within the data set as a comparison.

(2) Even if varicosities do not display features of classical synapses (for example, synaptic appositions), a lack of morphological characteristics is difficult to interpret in terms of release properties. The authors should specifically discuss that these non-synaptic varicosities may contain required protein release machinery and that even in the virtual absence of classical morphological features, such varicosities may release dopamine, with similar or distinct properties from the synaptic varicosities.

The full reviews are appended below to help clarify the concerns of individual reviewers. Each of the concerns should be addressed. Please note that none of the points require new experiments.

*Reviewer #1 (Recommendations for the authors):*

The introduction of the manuscript is way too excessive. Most of the introduction describes the results of the manuscript. The reviewer suggests shortening it and focus more on the background that is essential to place the current work in context.

The demonstration of the visualization of the mitochondria localized Apex2 is convincing in Figure 1 using classical TEM. It would be also nice to have examples in figure2 that contain labelled mitochondria.

The functional implication of the finding of the likelihood of having similar types of varicosities on a single axon is elusive.

*Reviewer #2 (Recommendations for the authors):*

This is a nice paper and I strongly support its publication in *eLife*. My specific comments relate to data analyses and presentation. I think that no new experiments are needed, but that limited additional analyses and tweaks in data presentation will improve the paper.

– Do the authors have an estimate of their labeling efficiency – are all DA neurons/axons labeled or only a subset? This could be important if a specific subset of axons is more efficiently labeled. A brief discussion of this should be included.

– In recent years, there has been a push to move away from glutaraldehyde fixation, but to instead use high pressure freezing followed by freeze substitution. This could affect the judgment of extracellular space as this space tends to collapse in glutaraldehyde fixed samples (see https://doi.org/10.7554/*eLife*.05793.001 for an example). A brief discussion on how this could affect the results (in the limitations section of the discussion) should be included. It could affect the judgement of cell-cell association, vesicle shapes, etc.

– One very important finding is that 98% of the dopamine varicosities do not form synaptic contacts. It is important to exclude that this very low number is an imaging artifact (I do not believe it is). Using the existing data set to reconstruct non-DA axons and to assess how many of their boutons with vesicles are associated with classical postsynaptic structures would strengthen this conclusion significantly if it is found that non-DA boutons indeed have a very high synaptic association judged by the same criteria. Synaptic vs non-synaptic appositions should be judged by a blinded experimenter.

– Related to this point: the results contain a brief statement on how synaptic vs. non-synaptic varicosities were defined, but an unequivocal description on how these types were defined is missing from the methods. Such a description has to be included.

– The paper refers mostly to vesicle size. A specific discussion of the electron density of the vesicle lumen should be included, because there is a large body of literature on catecholamine release from large dense core vesicles in chromaffin cells. It appears that the large vesicles do not have a dense core from the provided example images. This is an important point to discuss.

– Figure 4: I would recommend to express both in the caption and in the figure itself (for example with a pie chart) how rare the synaptic contacts are. This is a key finding that should be highlighted in a figure.

– Figure 6 and related data discussion: the cocaine data are largely derived from a lower resolution dataset with cytosolic DAB. It would be nice to provide images in this figure of axons and spinules (which are only provided for the higher resolution mito DAB dataset) to give the reader a sense for how spinules can be assessed with this method.

– For the cocaine experiment, it is concluded that cocaine mostly changes axon structure, but not individual varicosities. An important difference between the initial assessment of varicosities (Figures 1-5) and the cocaine experiments (Figures 6-9) is that they were performed with mito-DAB and cyto-DAB, respectively. Cyto-DAB may not be well suited to study individual varicosities, their vesicle contents, etc, and hence could confound this conclusion. The discussion should be more balanced in this respect and this specific point should be clearly stated.

– The cited cre lines in results (Backman, 2006) and methods (JAX 020080 = Zhuang 2005) are not the same. It is important to clearly state what cre line was used, and what the limitations of the respective cre line are.

– Most of the EM images have an overlayed color code. It would be nice to show each image side by side with the raw image (without color code) such that the quality of the EM can be easily assessed.

– Figure 9 and related text: while the result section is clear, the figure is not understandable because the key color code is missing from figure and legend. One can guess from Figure S9 what the color-code is (including the darker red). It should be made clear in the figure.

– Abstract: the current version is >330 words long, much beyond the *eLife* guidelines (150-200). It it is essential that the near-final abstract is carefully reviewed as most paper citations are (unfortunately) made for statements in the abstract.

– There is a data availability statement that outlines that data will be made available upon request. It would be nice to make this dataset fully available by the time of publication.

– In the result section that refers to Figure 6, Figure 5 is cited a few times. This is probably an error and Figure 6 should be cited (it would be helpful to add page and/or line numbers to the manuscript file such that it would be easy to refer to specific points in the text).

*Reviewer #3 (Recommendations for the authors):*

1) Abstract – 2nd page, line 1. It is questionable to refer to the prevalence with which DA varicosity form spinule-like structures as "frequent" based on quantified value of 15%.

2) Figure 2. The distribution of the number of vesicles of each type – small or large – associated with each category should be shown (except for varicosities with no visible vesicle). This will make clearer the degree of variations.

3) Figure 3A. It seems that the inter-varicosity distance should be considered also, as this could potentially affect the accumulation of vesicles. What was the average distance between varicosities and how variable were the distances? Was there any dependency on the distance between the adjacent varicosities and the type of vesicles contained within?

4) Figure 3. It may be premature to conclude that the DA axon can be classified according to the type of varicosity it tends to possess based on the current analysis. Could the authors exclude the possibility that nearby cells might influence the varicosity type within a certain contiguous domain?

5) With respect to expected morphological changes with cocaine exposure, could the authors introduce the specifics of the changes reported previously (cf. Beeler et al., 2009; Li et al., 2004; Singer et al., 2009)?

6) Figure 6. In panel A, the total length of representative axons shown for control vs. cocaine is drastically different, which is a source of concern. Also panel C suggests much shorter axon lengths are represented in the control sample. Could one estimate the number of branch points in reference to what could be labelled as a main axon, and follow the branch points according to main, primary, secondary, … branches? In that case, how do main axon lengths compare?

7) p.13, end of first paragraph: The analysis of varicosity type is lacking in cocaine-treated samples. It is not clear how the analysis of contact points could lead to the claim that "the nature of individual varicosities and interactions with target neurons remained similar".

8) Figure 7. Given that mitochondrial accumulation is a feature of the axonal swellings associated with cocaine treatment, it would be informative to know the quantification of mitochondria in the four types of varicosities identified in controls and in cocaine treated mice.

9) Figure 8. Glial association is observational, and causality to the presumed axonal remodeling remains highly speculative, at least from the data shown.

10) Figure 9C. The width of mitochondrial length distribution in control cannot be well appreciated by the overlapping bar histograms. Also, was there a change in mitochondrial density associated with its increase in length? An increase without a change in its density could support an increased demand for energy supply as what might be the case for extensive structural remodeling.

---

## [Author Response]

Essential revisions:All three reviewers acknowledge the valuable nature of the present EM data of DA axons. However, two major concerns have been raised regarding the interpretation of axonal varicosities.(1) An unequivocal definition for the classification of synaptic vs. non-synaptic varicosities is needed.

We agree that we should provide a more detailed description of varicosities and how we determine if they were synaptic or not. We have now added detailed text to the methods section:

“We defined boutons or varicosities, here used interchangeably, as large abrupt swellings along the length of axons, (i.e., the diameter of the axon increases ~ 2-fold or more in ~ 1µm of distance or less), consistent with previous classifications (Bodian, 1970). We classified whether varicosities were synaptic if we found ultrastructural evidence for: synaptic vesicles contained in the varicosity, if the vesicles are clustered, clear sign of a post-synaptic density, parallel pre- and post-synaptic membranes, and presynaptic active zones. We classified varicosities as synaptic if we could find evidence for 2 or more of the above, preferably one presynaptic measure and one post, (e.g., synaptic vesicles clustered opposed to a strong PSD), in any object with membrane to membrane apposition anywhere along the defined varicosity.”

Also, in order to exclude the possibility that the sparsity of classical synapses is due to some imaging artefact, include reconstructions of non-DA axons within the data set as a comparison.

We appreciate the reviewers' comments. We have now added a supplementary figure, Figure 4—figure supplement 3, where we show serial sections through 3 representative single synapses visualized by this approach (p. 62-63). These synapses were captured from the exact same dataset we analyzed mito-Apex2 DA boutons for classical ultrastructural signs of synapses.

Additionally, we are confident that we can detect ionotropic synapses. We, and others, have used this approach of SEM imaging of ultra-thin EM sections for analyzing neuronal connections in multiple publications. (Kasthuri, et al., Cell, 2015, Sampathkumar, et al., EJN, 2021, Sampathkumar, et al., PNAS, 2021, Morgan, Cell, 2016, Morgan and Lichtman, Neuron, 2020, Wildenberg, et al., Cell Reports 2021, Shapson-Coe, et al., BioRxiv, 2021) and with similar resolutions using blockface SEM approaches (Helmstaedter, et al., Nature 2013, Ding, et al., Nature 2016).

(2) Even if varicosities do not display features of classical synapses (for example, synaptic appositions), a lack of morphological characteristics is difficult to interpret in terms of release properties. The authors should specifically discuss that these non-synaptic varicosities may contain required protein release machinery and that even in the virtual absence of classical morphological features, such varicosities may release dopamine, with similar or distinct properties from the synaptic varicosities.

We thank the reviewers and editors for emphasizing this important distinction. We have added text to our Discussion section:

“We define here ‘synapse’ in the classical anatomical sense with ultrastructural evidence of presynaptic vesicles and postsynaptic specializations, (e.g., parallel membranes or a PSD). This definition leverages decades of combined EM and light microscopic work correlating EM signatures with molecular machinery (e.g., active zones contain presynaptic ca^2+^ channels and PSDs contains synaptic receptors). This work concludes primarily that most DA boutons show little evidence of classic EM signatures of synapses. It remains possible that molecular release machinery exists at these boutons without ultrastructural synaptic specialization, perhaps a fundamentally different type of NT release than classic inotropic transmission. Indeed, there is some evidence that DA axons contain release machinery, and, interestingly, that this machinery (i.e., RIM, ELK5) is only found in ~30% of dopamine varicosities (Liu *et al.*, 2018) further highlighting the complex nature to dopamine release. Thus, our results are a steppingstone to future correlated molecular and EM studies leveraging a range of new protocols (Fulton and Briggman, 2021; Micheva and Smith, 2007) that investigate how the presence of molecular machinery correlate with EM ultrastructure including different varicosity types we describe.”

The full reviews are appended below to help clarify the concerns of individual reviewers. Each of the concerns should be addressed. Please note that none of the points require new experiments.Reviewer #1 (Recommendations for the authors):The introduction of the manuscript is way too excessive. Most of the introduction describes the results of the manuscript. The reviewer suggests shortening it and focus more on the background that is essential to place the current work in context.

Thank you for pointing this out. We have substantially shortened the introduction emphasizing the background relevant for the work. We are happy to edit further. Please see p. 2-4.

The demonstration of the visualization of the mitochondria localized Apex2 is convincing in Figure 1 using classical TEM. It would be also nice to have examples in figure2 that contain labelled mitochondria.

We appreciate the suggestion and replaced the original images with varicosities of each vesicle class containing labeled Apex2+ mitochondria in Figure 2. (p. 48-49).

The functional implication of the finding of the likelihood of having similar types of varicosities on a single axon is elusive.

We thank the reviewer for pointing out this gap and added text that elaborates further on potential functional implications in our discussion. Please see p. 21-22:

“The nonrandom distribution of varicosity types in DA axons suggests that the release of potentially different types of transmitters is a property of individual DA axons and that DA axons might be further functionally classified by predominant varicosity type. Future experiments that, for instance, purify different vesicle sizes and identify their molecular contents could shed additional light on what these different classes functionally represent.”

Reviewer #2 (Recommendations for the authors):This is a nice paper and I strongly support its publication in eLife. My specific comments relate to data analyses and presentation. I think that no new experiments are needed, but that limited additional analyses and tweaks in data presentation will improve the paper.– Do the authors have an estimate of their labeling efficiency – are all DA neurons/axons labeled or only a subset? This could be important if a specific subset of axons is more efficiently labeled. A brief discussion of this should be included.

We thank the reviewer for pointing out this potential limitation to our study. We expand upon the text in our “Limitations” subheading to highlight multiple points mentioned in the reviews. In this expanded Limitations discussion we include limitations about

1. Labeling efficiency and labeling of only subsets of DA neurons (pg. 17-18):

“Second, while we use well documented transgenic lines that putatively targets all DA cell types indiscriminately within the VTA, we cannot rule out whether some types of axons are more efficiently labeled in our approach, (e.g., preferential expression based on AAV serotype or variability in surgical delivery of AAV). We believe these caveats are limited: we were careful to use the same batch of AAV9 virus for all experiments, we visually inspected the DAB/Apex2 reaction to ensure that the labeling intensity was approximately the same, and we also did not observe any behavioral or locomotor defects suggesting that toxicity was not an issue. Absolute labeling inefficiencies ought not to impact our characterization of dopamine axons (i.e., Figures 1-5) and the reproducibility of our results across animals in our cocaine studies suggests that absolute labeling inefficiencies were not so poor as to lead to confounds in statistical differences across individuals. It remains a possibility that sub-classes of DAT+ neurons (e.g., different co-transmitting DA neurons, or ones that project to different regions) might have different morphological features and responses to cocaine and thus we cannot exclude the possibility that the results presented here would not be universally applicable to all DA axons. We started these investigations as a necessary first step to evaluate expression, evaluate staining, and explore how connectomic datasets could be used to detail DA circuits. Future experiments where different DA cell types are deliberately differentially labeled (e.g., mitochondria and cytoplasmic-Apex2 targeted to two DA subtypes) in the same animal would further refine these initial results. Indeed, labeling of many potential DA cell types may allow us to further refine our classification system based on the composition of varicosities in individual axons (Figure 2-3).”

Discussion about how artifacts from aldehyde fixation could affect these results, (p.19-20):

“There are also possible confounds to our aldehyde fixation method. Aldehyde fixation, while commonly used for fixing large pieces of tissue (>300 µm), introduces fixation artifacts including distortion of cell morphology and loss of extracellular space (Korogod et al., 2015; Tsang *et al.*, 2018) which could potentially affect our evaluation of cell-cell associations and vesicle morphology. We believe the concerns about aldehyde fixation are mitigated for several reasons. First, we find that conventional synapses (e.g., glutamatergic spine synapses) prepared in the same fashion are easily detectable with classic ultrastructural signs (i.e., large vesicle cloud, post-synaptic density) (Figure 4—figure supplement 3). Thus, it is unlikely we missed, for instance, dopamine varicosities forming synapses or that empty varicosities were due to artifacts of aldehyde fixation. Secondly, for comparisons across control and cocaine treated mice, it is unlikely that aldehydes introduced any artifacts to our reported changes in branch number, mitochondrial length, and formation of retraction bulbs, especially given the fact that both experimental and control groups were fixed and prepared for EM on the exact same day, under the exact same conditions. Moreover, we did not make any quantitative comparisons of morphologies known to be affected by aldehyde fixation including loss of extracellular space, and proximity of glia to blood vessels and synapses (Korogod *et al.*, 2015). However, we cannot rule out that aldehyde fixation somewhat confound our observations on variable vesicle morphology or the 3D nature of spinules. Until recently, it was difficult to combine peroxidase staining with high pressure freezing (HPF), a fixation method with superior ultrastructural preservation (Korogod *et al.*, 2015), but recent protocols (Korogod *et al.*, 2015; Tsang *et al.*, 2018) suggest new experiments for leveraging HPF to more accurately reconstruct such cellular features in future experiments.”

– In recent years, there has been a push to move away from glutaraldehyde fixation, but to instead use high pressure freezing followed by freeze substitution. This could affect the judgment of extracellular space as this space tends to collapse in glutaraldehyde fixed samples (see https://doi.org/10.7554/eLife.05793.001 for an example). A brief discussion on how this could affect the results (in the limitations section of the discussion) should be included. It could affect the judgement of cell-cell association, vesicle shapes, etc.

See above and p. 19-20 of manuscript.

– One very important finding is that 98% of the dopamine varicosities do not form synaptic contacts. It is important to exclude that this very low number is an imaging artifact (I do not believe it is). Using the existing data set to reconstruct non-DA axons and to assess how many of their boutons with vesicles are associated with classical postsynaptic structures would strengthen this conclusion significantly if it is found that non-DA boutons indeed have a very high synaptic association judged by the same criteria.

This is an excellent point and suggestion to include analysis of non-DA axons. We performed exactly the test suggested by the reviewer and found overwhelming evidence that nearly all unlabeled boutons with vesicles in the same data sets could associated with classical postsynaptic structures. We have added the new result on p.9 and reproduced below. We also included a Figure 4—figure supplement 3 (p. 62-63) for examples of non-DA axons synapsing with dendritic spines for comparison to DA axon varicosities and putative synapses.

Finally, we examined whether unlabeled boutons, presumably from glutamatergic excitatory axons, also showed clear ultrastructural evidence of synapses. We returned to the same mito-Apex2 dataset and identified 100 axonal boutons containing a large vesicle cloud (i.e., 50 or more small vesicles) and asked whether we could identify an obvious PSD in membrane-to-membrane apposition to the bouton. We found that 100/100 pre-synaptic boutons were in close apposition to a post-synaptic dendritic spine or shaft with a stereotypically dark PSD staining, thus mitigating the possibility that the lack of observed DA synapses is due to our sample preparation or imaging approach.

Synaptic vs non-synaptic appositions should be judged by a blinded experimenter.

We thank the reviewers for this suggestion. We had another expert EM annotator re-examine boutons along labeled DA axons, identifying 25. The expert found, again, that the majority (24/25) of boutons made no signs of classical synapses (now defined more clearly in the Methods section). We have now added this verification to our Methods section:

“Because of the rarity of putative DA synapses, we had a third expert EM annotator re-examine 25 randomly chosen annotated boutons along mito-Apex2 labeled DA axons and found 24/25 of the boutons made no signs of classical synapses.”

– Related to this point: the results contain a brief statement on how synaptic vs. non-synaptic varicosities were defined, but an unequivocal description on how these types were defined is missing from the methods. Such a description has to be included.

We apologize for the oversight and have included a more detailed description in Methods:

“We defined boutons or varicosities, here used interchangeably, as large abrupt swellings along the length of axons, (i.e., the diameter of the axon increases ~ 2-fold or more in ~ 1µm of distance or less), consistent with previous classifications (Bodian, 1970). We classified whether varicosities were synaptic if we found ultrastructural evidence for: synaptic vesicles contained in the varicosity, if the vesicles are clustered, clear sign of a post-synaptic density, parallel pre- and post-synaptic membranes, and presynaptic active zones. We classified varicosities as synaptic if we could find evidence for 2 or more of the above, preferably one presynaptic measure and one post, (e.g., synaptic vesicles clustered opposed to a strong PSD), in any object with membrane to membrane apposition anywhere along the defined varicosity.”

– The paper refers mostly to vesicle size. A specific discussion of the electron density of the vesicle lumen should be included, because there is a large body of literature on catecholamine release from large dense core vesicles in chromaffin cells. It appears that the large vesicles do not have a dense core from the provided example images. This is an important point to discuss.

We thank the reviewer for pointing out this important distinction and we have added text to the discussion:

“We did not see the electron densities commonly associated with LCV, for example in the release of catecholamines (Grabner et al., 2005; Stevens et al., 2011), suggesting that the LCV seen in DA release might operate by different molecular release mechanisms (Edwards, 1998).”

– Figure 4: I would recommend to express both in the caption and in the figure itself (for example with a pie chart) how rare the synaptic contacts are. This is a key finding that should be highlighted in a figure.

Thank you for noting this, and its importance. We have added detail to figure 4 per the reviewer’s suggestions (p. 56-57).

– Figure 6 and related data discussion: the cocaine data are largely derived from a lower resolution dataset with cytosolic DAB. It would be nice to provide images in this figure of axons and spinules (which are only provided for the higher resolution mito DAB dataset) to give the reader a sense for how spinules can be assessed with this method.

We agree this will add to the value and transparency of the paper and have included lower resolution examples in Figure 6 along with more detail as a figure supplement (Figure 6—figure supplement 2). (p. 68-69, and p.72-73).

– For the cocaine experiment, it is concluded that cocaine mostly changes axon structure, but not individual varicosities. An important difference between the initial assessment of varicosities (Figures 1-5) and the cocaine experiments (Figures 6-9) is that they were performed with mito-DAB and cyto-DAB, respectively. Cyto-DAB may not be well suited to study individual varicosities, their vesicle contents, etc, and hence could confound this conclusion. The discussion should be more balanced in this respect and this specific point should be clearly stated.

We agree and have modified the results (pg. 12) to limit our interpretation to changes that are unambiguously identifiable using cyto-apex (i.e., contact points, branching) and we also add representative low res EM images to Figure 6 (p. 68-69) and as a supplement (Figure 6- Figure Supp 2; p.72-73) to demonstrate the visibility to contact points in cyto-apex datasets. We also add text to the summary to discuss limitations to cyto-apex.:

p.12:

“These results suggest that while as axons make new branches, they also form new contact points with surrounding neurons.”

p.20:

“Finally, we used cyto-Apex2 for reconstructing axons and their contact points for a several reasons. First, we found that tracing axons in low resolution EM data sets, for both controls and experimental groups, was substantially easier in cytoplasmic Apex2 axons, thus increasing our tracing throughput. In addition, we found that contact points, (e.g., spinules), were also easier to detect. Detecting either a darkly Apex2 labeled cytoplasmic process in an unlabeled structure, or vice versa, was easier because of the stark contrast between Apex2 labeled and unlabeled processes. In Figure 6 and Figure 6-figure supplement 2 an example of this difference is shown. While it is possible that cytoplasmic Apex2 expression could potentially obscure the internal contents of varicosities (e.g., synaptic vesicles or endoplasmic reticulum), we found little evidence that cyto-Apex2 obscured the relevant ultra-structural features that were investigated here.”

– The cited cre lines in results (Backman, 2006) and methods (JAX 020080 = Zhuang 2005) are not the same. It is important to clearly state what cre line was used, and what the limitations of the respective cre line are.

Thank you for catching this typo. We have corrected the reference and added text discussing the limitations to this chosen cre line:

“The mouse line used is a DAT-CRE knock-in strain. As a result, it disrupts one copy of DAT. Losing one copy of DAT is known to slightly reduce DA reuptake but heterozygote mice do not display any other phenotypes (Giros et al., 1996). In our cocaine experiments, the cocaine and saline groups have the same genotype to ensure comparisons are not confounded by strain differences.”

– Most of the EM images have an overlayed color code. It would be nice to show each image side by side with the raw image (without color code) such that the quality of the EM can be easily assessed.

Thank you for pointing out that the color overlay could affect how well the reader sees the raw data. We feel for non-EM experts color coding helps direct the reader’s attention but worry that adding the image side-by-side could be confusing and make the figures too big. As an alternative we make the color coding more transparent so that more of the raw image is visible. We have also added significantly more examples of EM images as figure supplements with arrows pointing to relevant objects rather than color coding. We hope this is a satisfactory compromise.

– Figure 9 and related text: while the result section is clear, the figure is not understandable because the key color code is missing from figure and legend. One can guess from Figure S9 what the color-code is (including the darker red). It should be made clear in the figure.

We apologize for the confusion and missing key. We have added this detail to figure 9 (p. 80).

– Abstract: the current version is >330 words long, much beyond the eLife guidelines (150-200). It it is essential that the near-final abstract is carefully reviewed as most paper citations are (unfortunately) made for statements in the abstract.

We have shortened the abstract to 201 words.

– There is a data availability statement that outlines that data will be made available upon request. It would be nice to make this dataset fully available by the time of publication.

We completely understand and have full intent to do so. We are working to find the best public database with the front runner being neurodata.io. Because each dataset is ~300-400 Gb and there are 6 datasets, and the data has to be converted into usable formats (e.g., neuroglancer), neurodata.io would like the data to be accepted for publication first.

– In the result section that refers to Figure 6, Figure 5 is cited a few times. This is probably an error and Figure 6 should be cited (it would be helpful to add page and/or line numbers to the manuscript file such that it would be easy to refer to specific points in the text).

We have corrected the text to point to the correct figure.

Reviewer #3 (Recommendations for the authors):1) Abstract – 2nd page, line 1. It is questionable to refer to the prevalence with which DA varicosity form spinule-like structures as "frequent" based on quantified value of 15%.

We agree this is confusing and have changed the text to more accurately represent the data:

“DA axon varicosities rarely make specific synapses (<2%, 6/410), but instead are more likely to form spinule-like structures (15%, 61/410) with neighboring neurons.”

2) Figure 2. The distribution of the number of vesicles of each type – small or large – associated with each category should be shown (except for varicosities with no visible vesicle). This will make clearer the degree of variations.

We thank the reviewer for this suggestion. We calculated the distribution of number of vesicles for each type and for each type of vesicle, in Author response table 1. We will add this data to the Results section of the manuscript (p.7).

**Author response table 1. sa2table1:** 

Varicosity type	Average no	SEM
small	30.89583333	3.2
large	9.222222222	0.8
large/Small:Total	35.07894737	5.5
large/small:large	6.921052632	0.9
large/small:large	28.15789474	4.8

3) Figure 3A. It seems that the inter-varicosity distance should be considered also, as this could potentially affect the accumulation of vesicles. What was the average distance between varicosities and how variable were the distances? Was there any dependency on the distance between the adjacent varicosities and the type of vesicles contained within?

We thank the reviewer for suggesting these analyses. We found that the mean distance between adjacent varicosities (n=170 varicosities over 29 axons) was ~5.3 μm with a standard deviation of 4.1 um. The histogram of distances between adjacent varicosities was well fit by a lognormal distribution with mu = 8.37204 [8.27752, 8.46656] σ = 0.624272 [0.56422, 0.698742], with confidence intervals indicated in brackets. We did not see any obvious sign that either extremely short or long distances between adjacent varicosities correlated with the types of varicosities. We have now added these results to the main text (p.8), and reproduced below, and added the resulting histogram and lognormal fit to Figure 3- Figure Supp 2 (p.56-57).

“We next asked whether the inter-varicosity distance was correlated with different varicosity types as a potential clue for varicosity types being organized along an axon depending on their vesicle contents. We found the distances between pairs of varicosities (n = 170 varicosities across 29 DA axons) fit a log normal distribution (Figure 3 —figure supplement 2A) without any obvious signs of extremely long or short inter-varicosity distances clustering to a particular varicosity type.”

4) Figure 3. It may be premature to conclude that the DA axon can be classified according to the type of varicosity it tends to possess based on the current analysis. Could the authors exclude the possibility that nearby cells might influence the varicosity type within a certain contiguous domain?

We believe this is an important point and we cannot currently eliminate the possibility that varicosity type is correlated to nearby cells or processes. We believe such proximity analyses is best done in the context of knowing which cells/processes express dopamine receptors, e.g. future studies that combine molecular labeling with large volume EM. This additional labeling of processes that contain DA receptors will be critical in defining the spatial extent, e.g. ‘nearby’, over which such analyses should be performed. We now add this interesting point to the discussion:

“Finally, an intriguing possibility is that nearby cells might influence the varicosity type found on individual DA axons. Such proximity analyses are likely best done in the context of knowing which cells/processes express dopamine receptors (e.g., future studies that combine molecular labeling with large volume EM (Fulton and Briggman, 2021; Micheva and Smith, 2007)). This additional labeling of processes that contain DA receptors will be critical in defining the spatial extent (e.g., ‘nearby’) over which such analyses should be performed.”

5) With respect to expected morphological changes with cocaine exposure, could the authors introduce the specifics of the changes reported previously (cf. Beeler et al., 2009; Li et al., 2004; Singer et al., 2009)?

We have included these references, p. 10.

6) Figure 6. In panel A, the total length of representative axons shown for control vs. cocaine is drastically different, which is a source of concern.

We apologize for this confusion in 6A and agree this can be misleading. We now show control axons in 6A that are similar in length to cocaine exposure, p. 68.

Also panel C suggests much shorter axon lengths are represented in the control sample.

We thank the reviewer for this suggestion We returned to our datasets to trace more axons in control mice. We have added the additional data to panel C (p.68, and see Author response image 1) .

**Author response image 1. sa2fig1:** Original plot (left) and updated with more data collected (right).

Could one estimate the number of branch points in reference to what could be labelled as a main axon, and follow the branch points according to main, primary, secondary, … branches? In that case, how do main axon lengths compare?

While we appreciate that identifying primary and secondary branches of axons would be valuable, we are not confident that we can do so in these datasets. We do see small differences in axonal caliber, often associated with main vs. secondary branches but given that the region examined is likely where axons demyelinate and ramify, we are unsure about how such caliber changes correlate with location on the axonal arbor. Thus, we prefer to avoid these analyses for clarity. We are happy to revisit this question if such analyses are critical.

7) p.13, end of first paragraph: The analysis of varicosity type is lacking in cocaine-treated samples. It is not clear how the analysis of contact points could lead to the claim that "the nature of individual varicosities and interactions with target neurons remained similar".

We apologize for the including claims about varicosity type and have modified the text to address contact points only (p.12).

8) Figure 7. Given that mitochondrial accumulation is a feature of the axonal swellings associated with cocaine treatment, it would be informative to know the quantification of mitochondria in the four types of varicosities identified in controls and in cocaine treated mice.

We have included a more detailed description of how mitochondria were quantified in the methods section (p.44) and refer readers to the methods in the main body of the manuscript (p.14). We thank the reviewer for this suggested analysis. On reinspection we find that all four types of varicosities contain mitochondria at similar percentages (~50%. We have added a new supp Figure 3- Figure Supp 2B; p. 54-55) to detail this result and have now added text (p. 8-9) to emphasize the result:

“Finally, we scored for the presence or absence of mitochondria in each varicosity type to further ascertain whether mitochondria were uniquely associated with certain varicosity types. However, we found that for each varicosity type, there was a ~50% chance that a mitochondrion was also present, eliminating any obvious association between mitochondria location and varicosity type (Figure 3 —figure supplement 2B).”

9) Figure 8. Glial association is observational, and causality to the presumed axonal remodeling remains highly speculative, at least from the data shown.

We agree these results are qualitative, and have added text to the discussion that states this caveat explicitly with suggestions on future experiments more specifically focused on glia (p.27):

“Because our sample size is limited to qualitative observations, future experiments that manipulate the presence or activity of glia (De Luca et al., 2020; Nguyen et al., 2020; Paolicelli et al., 2017) will provide potentially causal roles glia have in DA axons remodeling after cocaine exposure.”

10) Figure 9C. The width of mitochondrial length distribution in control cannot be well appreciated by the overlapping bar histograms. Also, was there a change in mitochondrial density associated with its increase in length? An increase without a change in its density could support an increased demand for energy supply as what might be the case for extensive structural remodeling.

We apologize for the lack of clarity and have adjusted the histogram to make it clearer by breaking the saline and cocaine datasets into two histograms (p.80). For density, we see no change in the number of mitochondria/unit of axon length (Figure 9—figure supplement 1, p. 83-84). We further clarified in the text with mean and sem of mito/length (p.15) and see below. We have also added to the results that mitochondria increase in length but no change in density (p.15). Finally, we agree with the reviewer that changes in mito length without changes in mito density have important implications for energy demand. However, we could not find sufficient precedent in the literature to make a conclusive discussion point.

Control: 0.14 mitochondria/µm mean, ± 0.02 SEM

Cocaine: 0.16 mitochondria/µm mean, ± 0.02 SEM